# Field Testing of Selected Salt-Tolerant Screened Balsam Poplar (*Populus balsamifera* L.) Clones for Use in Reclamation around End-Pit Lakes Associated with Bitumen Extraction in Northern Alberta

**Yue Hu** [1,*] **, David Kamelchuk** [2]**, Richard Krygier** [3] **and Barb R. Thomas** [1] 

1    Department of Renewable Resources, University of Alberta, Edmonton, AB T6G 2E3, Canada; bthomas@ualberta.ca
2    Alberta-Pacific Forest Industries Inc., Boyle, AB T0A 0M0, Canada; david.kamelchuk@alpac.ca
3    Natural Resources Canada, Canadian Forest Service, Canadian Wood Fibre Centre, Edmonton, AB T6H 3S5, Canada; richard.krygier@canada.ca
*    Correspondence: yhu6@ualberta.ca

**Abstract:** For the oil sands mine sites in northern Alberta, the presence of salty process affected water, a byproduct of the hot-water bitumen extraction process, is anticipated to pose a challenge on some reconstructed landforms. The fundamental challenge when re-vegetating these sites is to ensure not only survival, but vigorous growth where plants are subjected to conditions of high electrical conductivity owing to salts in process affected water that may be contained in the substrate. Finding plants suitable for high salt conditions has offered the opportunity for Alberta-Pacific Forest Industries Inc. (Al-Pac) to investigate the potential role of using native balsam poplar (*Populus balsamifera* L.) as a key reclamation species for the oil sands region. Two years of greenhouse screening (2012 and 2013) of 222 balsam poplar clones from Al-Pac's balsam poplar tree improvement program, using process affected discharge water from an oil sands processing facility in Ft. McMurray, has suggested an opportunity to select genetically suitable native clones of balsam poplar for use in reclamation of challenging sites affected by process water. In consideration of the results from both greenhouse and field testing, there is an opportunity to select genetically suitable native clones of balsam poplar that are tolerant to challenging growing conditions, making them more suitable for planting on saline sites.

**Keywords:** oil sands reclamation; end-pit lake; balsam poplar; salt tolerance

## 1. Introduction

In Canada, the Alberta oil sands region is located in the Cold Lake, Peace River, and Athabasca regions in the North America Boreal Plain and covers approximately 142,200 km$^2$ [1]. Currently, approximately 856,000 barrels of bitumen per day (bbl day$^{-1}$) are produced in the mineable portion of the Athabasca region [1]. Surface mining for oil sands production in the Athabasca Oil Sands Region (AOSR) of Alberta has resulted in a cumulative disturbance footprint of 895 km$^2$ (until 2013), with only 0.2% of the total land base disturbed by mining being certified as reclaimed by the Government of Alberta [2]. In addition, open-pit mining leaves a reconstructed landscape of overburden dumps and tailings deposits that require reclamation, targeting self-sustaining and locally common ecosystems [3]. The process of bitumen extraction requires vast amounts of water [4] and the resultant oil sands process water (OSPW) must be contained and not returned to the region's river system owing to a current zero-discharge policy by the Alberta Environmental Protection and Enhancement Act [5]. This requirement has resulted in over a billion cubic meters of tailings water being held through various types of containment systems [6], one of which is often referred to as an 'end-pit' lake. As development expands, large areas of

disturbed land in the oil sands region will require reclamation with suitable, well-chosen, native plant material. Reclamation, in the AOSR, is defined as the process to return the disturbed ecosystems to an "equivalent land capacity" as the pre-disturbance ecosystem [7]. This may involve practices such as recontouring the ground, replacing the subsoil and topsoil, revegetation, and monitoring the environmental conditions [7].

Soil salinity and sodicity in oil sands reclamation areas have been listed among the most challenging revegetation concerns [8–10]. Salt stress leads to reductions in growth, productivity, and survival in numerous plant species [11]. The stress induced by salinity on plants is the result of three mechanisms: osmotic stress due to a more negative soil water potential, accumulation of toxic ions, and disturbances in nutrient balance [11]. These effects, in turn, lead to reductions in growth, productivity, and survival in numerous plant species [11]. In the AOSR in Alberta, salinity problems associated with OSPW and exposed marine shale overburden are two major potential challenges when reclaiming upland landscapes [9]. Salt-stress is particularly detrimental for boreal woody species as most exhibit relatively low tolerance to salinity [9]. The fundamental challenge when revegetating these sites is not only to ensure survival, but to achieve growth rates appropriate to the ecosystem class even where plants are subjected to conditions of high electrical conductivity (EC) owing to salts in process affected water that is contained in the substrate.

The challenge of finding plants suitable for high salt conditions has offered the opportunity for Alberta-Pacific Forest Industries Inc. (Al-Pac) to investigate the potential role of using native balsam poplar (*Populus balsamifera* L.) as a key reclamation species for the oil sands region. Poplars are used throughout North America to reclaim sites containing heavy metals, salts, pesticides, solvents, explosives, radionuclide hydrocarbons, and landfill leachates [12–14]. Many studies have suggested that *Populus* species are tolerant to salinity and can even lower soil salinity [15–18]. For example, Euphrates poplar (*Populus euphratica* Oliv.) can grow well in soils with up to 8000 mg $L^{-1}$ salinity [15]. Moreover, Liu et al. (2001) [19] reported that white poplar (*Populus alba* L.) tolerated 2000 mg $L^{-1}$ salinity irrigation in a sandy soil in a greenhouse for two years. Poplars are also well suited for phytoremediation thanks to their ability to uptake high levels of nutrients and mineral salts, accumulate above and below ground carbon, improve soil structure and function, and reduce erosion [12,16]. Balsam poplar is a desirable species for boreal forest reclamation thanks to its fast growth and ease of vegetative propagation [20], combined with its natural role as a pioneer species. In addition, the EC tolerance range for balsam poplar is very high, ranging from 14.58 to 31.38 mS $cm^{-1}$, whereas the tolerance of white spruce (*Picea glauca* (Moench) Voss) is 8.75 to 14.92 mS $cm^{-1}$ and that of jack pine (*Pinus banksiana* Lamb.) is even lower at 1.02–6.33 mS $cm^{-1}$ [9].

Al-Pac is a pulp company that manages a 6.37 million ha forest management agreement (FMA) area in northeastern Alberta with an overlapping tenure with the oil sands region of Alberta, Canada. Balsam poplar is native to the region and has been the focus of Al-Pac's controlled parentage (tree improvement) program (CPP) (PB1-Alberta-Pacific Controlled Parentage Program plan for balsam poplar (2011)). The CPP consists of clones selected from within the FMA area (also the CPP deployment region) and outside the FMA area, with a minimum of 10 clones per provenance and 52 provenances. Approximately 520 clones were selected and have been planted on six test sites throughout the FMA area, including extreme (i.e., dry) locations, to investigate both local adaptability and potential regional adaptability under climate change. While Al-Pac is testing these trees for their reforestation potential, they are also of significant interest for their oil sands reclamation potential. After two years of greenhouse screening (2012 and 2013) of 222 balsam poplar clones from Al-Pac's program and based on their responses to varying levels of exposure to OSPW, clones were grouped into three categories (see Section 2.2.1 for details) for further field testing.

Our objective for the greenhouse study was to identify clones through screening and select genotypes that would be expected to survive and grow when used for reclamation on sites affected by OSPW. For the field trial, our objective was to test and identify balsam

poplar clones, selected for salt tolerance from the greenhouse trials, exhibiting higher survival and increased growth (e.g., height and diameter) on reclamation sites compared with the following: (i) clones that did not exhibit tolerance to elevated salt levels in the greenhouse trials and (ii) a local seed zone Stream 1 wild balsam poplar cutting collection (local control).

## 2. Materials and Methods

### 2.1. Greenhouse Set-Up (2012&2013)

2.1.1. Plant Material and Growing Conditions

Two aeroponic greenhouse experiments were conducted in 2012 and 2013 using balsam poplar clones selected from Al-Pac's balsam poplar CPP. We selected 148 and then another 86 clones for screening in 2012 and 2013, respectively. In addition, 12 of the top clones from the 2012 experiment were retested in 2013.

All trees were propagated from 10 cm long dormant hardwood cuttings, from 1-year-old tissue, collected during the winter, prior to each experiment, from stooling beds grown at the Al-Pac mill site (54° 53′ N, 112° 51′ W, 575 m). Cuttings were stored in a chest freezer prior to commencement of the experiment. Cuttings were grown aeroponically in plastic containers filled with one of three treatment solutions and each aerated using a tubing system connected to a dedicated air compressor. The experiment was a completely randomized design with three treatments: (1) 100% reverse osmosis (RO) water (city water run through a reverse osmosis system); (2) 25% OSPW combined with 75% RO water; and (3) 50% OSPW with 50% RO water, with three replicates for each clone and water treatment combination (three containers per treatment) for a total of nine containers. Each treatment container held 80 L of solution.

The OSPW was collected directly from an outflow spout at a mine facility in Ft. McMurray, AB, into plastic jugs and transported to the Northern Forestry Centre (Natural Resources Canada, Canadian Forest Service) in Edmonton, AB, where the experiments were conducted each year. Prior to initiating the experiment, 170 mL of Hoagland's solution [21] was added to each treatment container. A near neutral pH was maintained 15 days prior to the start of the experiment and then monitored during the experiment for all water treatments by adding either phosphoric acid ($H_3PO_4$) (if higher than 7.5) or potassium hydroxide (KOH) (if lower than 6.5). Additional RO water was added to the containers at week four and week six to maintain adequate levels of liquid, compensating for water used by the plants and through evaporation. Additional OSPW was not added.

In 2012 and 2013, once pH was stabilized and after the experiment began (day 0), pH was maintained between 6.55 and 7.26 for all three water treatments for the duration of each experiment. The mean pH values for control (100% RO), 25% process water, and 50% process water were 6.91, 6.96, and 6.96, respectively, in 2012. In 2013, the mean pH values for control, 25% process water, and 50% process water were 6.96, 6.91, and 6.90, respectively. The mean electrical conductivity (EC) levels for control (100% RO), 25% process water, and 50% process water were 1.16 mS cm$^{-1}$, 2.14 mS cm$^{-1}$, and 3.28 mS cm$^{-1}$, respectively. In 2013, the mean EC levels for control, 25% process water, and 50% process water were 1.08 mS cm$^{-1}$, 2.25 mS cm$^{-1}$, and 3.31 mS cm$^{-1}$, respectively.

Containers (97 cm × 77 cm × 44 cm) used for this experiment had a cell arrangement of 11 cells long by 15 cells wide with a cell opening of 4 cm diameter into which a rubber bung, 4 cm long, was placed. Cuttings were placed into a hole in the middle of the rubber bung that fit into the cells in the lid of the container, suspending the cutting above the water. Cuttings were completely randomized for the location in each container. The experimental greenhouse had a day time temperature of approximately 24 °C and a mean night time temperature of 18 °C. Humidity was maintained at 65–85% with an 18 h photoperiod. In 2012, the cuttings were planted on 4 July (day 0) and grown until 17 August (day 44). In 2013, the cuttings were planted on 14 August (day 0) and grown until 18 October (day 65). An extended photoperiod was maintained with natural light supplemented with sodium vapor lamps at a light intensity of 250 μmol m$^{-2}$ s$^{-1}$. Each container had its own water

pump that sprayed the solution into the air space where the cuttings were suspended, and the roots grew, every 30 min. In 2012, two water pumps failed (one for treatment 2 and one for treatment 3) just prior to harvest, resulting in only two replicates for each treatment at harvest, and in 2013, one water pump failed for treatment 2 at day 30. All data were collected up until the point of pump failure, which was day 22 in the 2012 experiment; therefore, only the final harvest and gas exchange data were affected.

2.1.2. Data Collection

In 2012, water samples were collected from the containers and analyzed three times during each experiment. Treatment water was sampled prior to planting the cuttings (day 15), near the middle of the growth period (~day 35), and at the end of the experiment (day 44, 2012 and day 65, 2013) and analyzed for basic nutrients, pH, and EC by the Biogeochemical Analytical Service Laboratory at the University of Alberta. Initial cutting diameters (mm) were measured for all trees at planting. Mortality was assessed prior to harvesting. To be considered DEAD, the cuttings were required to have had leaves emerge and then die; otherwise, a cutting that never flushed was considered a missing value. At the end of the experiment, prior to measurements and destructive sampling, a qualitative visual assessment of tree health was completed using the following scale: (1) dead tree; (2) tree was dying, leaves or stem were wilting and turning black; (3) tree appeared stressed, significant yellowing, or dropping of leaves; (4) tree showed signs of chlorosis, but otherwise looked healthy; (5) leaves were green and tree looked healthy; and (6) leaves were dark green and tree was thriving. Final height (cm) was measured from the base of the new growth (attachment of stem to cutting) or the rubber bung surface, whichever was higher, to the base of the terminal bud. Final basal stem diameter (mm) was measured at the base of the new growth (attachment of stem to cutting), or the rubber bung surface, whichever was higher. In 2012, photosynthesis rate (A) was measured using an infrared gas analyzer (LI-6400; Li-Cor, Lincoln, NE, USA) prior to the final harvest. Measurements were made on one fully expanded mature leaf per cutting between 08:00 and 13:30 with a supplied (saturating) light level of 1200 PAR. Following growth measurements, plants were destructively harvested and separated into leaf, stem, root, and original cutting components; these were oven-dried in paper bags for 72 h at 65 °C and then weighed and used to calculate biomass and root/shoot ratio.

*2.2. Field Testing (2014–2019)*

2.2.1. Treatment Groups

Thirty-five balsam poplar clones from the PB1-CPP were screened and assessed for salt tolerance according to the greenhouse study [22]. Twenty-five of the selected clones were the top performing clones in the 50% process affected water treatment (treatment 1) and were selected as the 'high salt tolerant' treatment group and 10 clones that were poor performing clones in the 50% process affected water treatment were selected as a 'low salt tolerant' control group (treatment 2). There was an additional Stream 1 vegetative control lot collected from the local seed zone (CM2.2) with a minimum of 75 genotypes [23], which was not screened previously in the greenhouse (treatment 3).

2.2.2. Plant Material

One-year-old whips were collected from the Al-Pac mill site in February 2014, processed into 10 cm long cuttings, and placed in freezer storage at −3 °C. Cuttings were removed from freezer storage and soaked for two days in cool, fresh water prior to striking into Beaver Plastics® 512A styroblock (Beaver Plastics Ltd., Acheson, AB, Canada) containers on 4 June 2014 at Bonnyville Forest Nursery. Stream 1 control cuttings were collected in the winter of 2013/14 and kept in freezer storage until being soaked for one day and struck into 512A styroblock containers on 13 June at Bonnyville Forest Nursery.

All cuttings were grown under commercial nursery growing conditions from June to September 2014. Once they were hardened off and set bud, rooted cuttings were sorted and labeled and transported to the mine site. Planting was completed by 15 October 2014.

### 2.2.3. Testing Environment

The end-pit lakeshore used for this study is located north of Ft. McMurray (57°0′30″ N, 111°37′18″ W, 290 m a.s.l.) (Figure A2). The climate is considered a "warm-summer humid continental climate (Dfb)" according to Köppen climate classification [24]. The 20-year average (1999–2019) mean annual precipitation is 474 mm and the average temperature is 1.8 °C [24]. The site has a gentle slope running parallel to the water's edge with good nutrient condition. The former 50 to 60 m deep mine pit was largely filled with fluid fine tails (FFTs), and then capped with 4 m of process water and later 2 m of fresh water [25,26]. Given that the pore water of the FFT and the process water on top are brackish, vegetation on the shore of the lake was expected to be subjected to salty water (roughly 10% of the salinity of seawater) for the foreseeable future.

### 2.2.4. Experiment Design

The trial was a randomized block design and was planted on 15 October 2014. There were four ramets of each of 35 Al-Pac clones (25 treatment 1; 10 treatment 2) and 60 Stream I control trees (treatment 3) planted in each of three blocks on the south shore of the end-pit lake (Figure A3). Each block contained a total of 200 trees. In order to reduce within block variability, blocks were laid out with five trees running perpendicular from the lakeshore by 40 trees parallel to the lakeshore. Trees were planted 1 m apart in rows moving away from the lakeshore, with block 1 as close to the edge of the water as possible to maximize exposure to potentially saline lake water and ground water discharge from the adjacent hillside. All blocks followed the curving edge of the lake to keep them at as consistent an elevation and soil moisture condition within each block as possible. Rows up from the lake were tilled and covered with plastic mulch prior to planting. These rows were spaced approximately 3 m apart. Additional plastic mulch was placed manually between the rows to cover the entire trial area to minimize weed competition. Tree locations and identities were individually marked and mapped (Figure A3).

### 2.2.5. Growth and Survival Data Collection

Tree height (Ht) and basal root collar diameter (RCD) were measured according to protocols described in the trial measurement manual [27]. All trees were measured after installation for Ht, RCD in year 1, and diameter at breast height (DBH) was measured starting in year 2, and they were remeasured each fall in years 1, 2, 3, 4, and 5 (fall 2019). Survival was evaluated based on a visual assessment of the above ground stem from 2015 to 2019.

### 2.2.6. Tissue Nutrient Analysis

In summer 2016, two sets of leaf tissue samples were collected. Two leaves were collected from each live tree of the 35 clones and grouped into composite samples by clone and block. The Stream 1 control treatment trees had two leaves per tree collected from six randomly selected trees from each block and grouped to provide a single composite sample. A total of 108 samples were collected for the primary sample analysis. The Stream 1 control trees were chosen for the heavy metal analysis owing to the composition of this lot (i.e., 60 trees/block, minimum of 75 clones collected in the lot) representing a random, composite sample of multiple clones and collected from trees not used for the primary tissue analysis.

All 108 tissue samples were analyzed at Exova Laboratories, Surrey, British Columbia, to determine the uptake of nutrients and other compounds from the site (including boron, calcium, copper, iron, magnesium, manganese, molybdenum, phosphorous, potassium, sodium, sulfur, zinc, and nitrogen). The six Stream 1 control tree samples were also

analyzed for 33 heavy metals (including aluminum, antimony, arsenic, barium, beryllium, bismuth, boron, cadmium, chromium, cobalt, copper, iron, lead, lithium, magnesium, manganese, mercury, molybdenum, nickel, phosphorus, potassium, selenium, silicon, silver, sodium, strontium, sulfur, thallium, tin, titanium, vanadium, zinc, and zirconium). The heavy metal analysis was completed as an indicator only of potential heavy metal accumulation.

*2.3. Data Analysis*

For both greenhouse and field studies, all growth and nutrient data were analyzed by two-way mixed model analysis of variance (ANOVA) using SAS 9.4 [28].

For the greenhouse study, treatment (0, 25%, or 50% OSPW) was considered a fixed effect and clone and container were considered random effects. The treatment × clone interaction was also included in the model and initial diameter of the cutting was used as a covariate. Following significant main effects analysis, multiple comparisons among means were completed using the Student–Newman–Keuls (SNK) test. We used $p \leq 0.05$ to determine significance.

For the field testing, treatments were considered a fixed effect and block considered as a random effect. Multiple comparisons among means were completed using the Student–Newman–Keuls test with $p \leq 0.05$ used to determine significance.

## 3. Results

*3.1. Greenhouse Study*

3.1.1. Survival and Growth

In both 2012 and 2013, there were significant effects of treatment ($p < 0.001$, 2012; $p < 0.001$, 2013) and clone ($p < 0.001$, 2012; $p = 0.036$, 2013) on visual health assessment. The average ratings of the visual health assessment for the 2012 experiment, done at the end of the experiment, were 3.84, 3.39, and 2.81 for the control, 25%, and 50% treatments, respectively, out of a maximum score of 6. The control was significantly healthier ($p < 0.05$) than those in either the 25% or 50% treatments, which also differed from one another ($p < 0.05$). Most of the trees in the 25% and 50% treatments showed signs of chlorosis in both the younger and older leaves, and between the leaf veins. Some trees that ranked as a 3 or lower had necrotic leaf spots, were losing leaves, or in the most severe cases were dead.

In the 2013 experiment, the average visual health assessment ratings were 4.48, 3.95, and 4.38 for the control, 25%, and 50% treatments, respectively. The control and 50% treatments were significantly healthier ($p < 0.05$) than the 25% treatment; however, there was no significant difference between control and 50%. Observation of mortality showed that overall survival for the control treatment was 77.0%, the 25% solution was 80.4%, and the 50% solution was 64.9%. Survival ranged from 0 to 100%.

In both 2012 and 2013, there were significant effects of treatment and clone for final stem height, stem basal diameter and stem, and root and leaf biomass, with no clone by treatment interaction effect (Table 1). In 2012, the overall mean stem height for all clones did not differ between the control and 25% OSPW treatment, measuring on average 26 cm. Mean stem height for the 50% treatment was 30% significantly lower than both the control and 25% treatment. In 2013, the overall mean stem height for all clones in the control treatment was significantly greater than those in either the 25% or 50% treatments, which also differed from one another.

For stem basal diameter in the 2012 experiment, there was a decreasing trend from the control to the 25% to 50% treatments, with means averaging about 3.40 mm (Table 1). The trees in the 50% treatment were significantly smaller than in the control and 25% treatments. In the 2013 experiment, mean stem basal diameter did not differ between the control and 25% treatment; however, stem basal diameter in the 50% treatment was significantly lower ($p < 0.001$) than either the control or 25% treatment. Initial cutting diameter was used as a covariate for both the final stem height and final basal diameter analyses; it was not

significant for the final stem height and final basal diameter in 2012 ($p = 0.06$; final height, $p = 0.052$; basal diameter), but it was significant in 2013 ($p = 0.034$; final height, $p = 0.003$; basal diameter).

**Table 1.** Mean values (±SE) for growth and biomass measurements of balsam poplar in 2012 (top) and 2013 (bottom) for all three treatments. Significant differences between treatment means within each row are indicated by different letters based on results of analysis of variance followed by post-hoc Student–Newman–Keuls (SNK) tests.

| 2012 | Treatment | | |
|---|---|---|---|
| | **Control** | **25% process water** | **50% process water** |
| Final stem height (cm) | 26.49 ± 0.73a | 26.74 ± 0.80a | 18.08 ± 0.62b |
| Stem basal diameter (mm) | 3.46 ± 0.06a | 3.40 ± 0.07a | 3.20 ± 0.06b |
| Stem biomass (g) | 0.37 ± 0.02a | 0.31 ± 0.02b | 0.16 ± 0.01c |
| Root biomass (g) | 0.19 ± 0.01a | 0.13 ± 0.01b | 0.09 ± 0.01c |
| Leaf biomass (g) | 0.86 ± 0.04a | 0.73 ± 0.04b | 0.52 ± 0.03c |
| Total biomass (g) | 1.42 ± 0.08a | 1.17 ± 0.06b | 0.77 ± 0.04c |
| **2013** | **Treatment** | | |
| | **Control** | **25% process water** | **50% process water** |
| Final stem height (cm) | 39.79 ± 1.28a | 33.70 ± 1.43b | 31.15 ± 0.98c |
| Stem basal diameter (mm) | 4.69 ± 0.09a | 4.62 ± 0.11a | 4.06 ± 0.07b |
| Stem biomass (g) | 1.06 ± 0.07a | 0.86 ± 0.07b | 0.54 ± 0.03c |
| Root biomass (g) | 0.60 ± 0.04a | 0.47 ± 0.04b | 0.30 ± 0.02c |
| Leaf biomass (g) | 1.63 ± 0.08a | 1.30 ± 0.09b | 1.04 ± 0.05c |
| Total biomass (g) | 3.32 ± 0.19a | 2.88 ± 0.21b | 1.90 ± 0.10c |

In the 2012 experiment, leaf, stem, root, and total biomass decreased significantly from the control to the 25% and 50% treatments (Table 1). In the 2012 experiment, the control treatment had the highest root/shoot ratio (0.15 ± 0.005) followed by the 50% treatment (0.14 ± 0.005), while the 25% treatment had the lowest root/shoot ratio (0.12 ± 0.006). There was no significant difference between the control and 50% treatment, although both differed from the 25% treatment. In the 2013 experiment, the control treatment had the highest root/shoot ratio (0.19 ± 0.008) followed by the 25% treatment (0.17 ± 0.008) and then the 50% treatment (0.15 ± 0.005).

Photosynthesis rates (A) were significantly influenced by both clone and treatment ($p < 0.001$) in 2012. Post-hoc comparisons ($p < 0.05$) showed that the control treatment had a significantly higher A than either the 25% or 50% process water treatments, which did not differ from one another. Pearson's correlation coefficient (r) between A and overall biomass was 0.435 for control, 0.434 for the 25% process water, and 0.435 for the 50% process water treatments. Significantly positive correlations ($p < 0.001$) were detected between all the treatments' total biomass and A. Despite some clones performing better under the OSPW treatments, the control water treatment plants had the highest rates of photosynthesis as well as the highest visual score for plant health at 3.84 in 2012 (vs. 3.39 in 25% and 2.81 in 50%).

Nitrogen and magnesium levels appeared to be adequate as compared with the control for all treatments (Table 2); however, there were noticeable decreasing trends with control > 25% > 50% process-affected water for iron, indicating that an iron deficiency may have been present. It was observed that the iron levels in the 25% and 50% process water samples were very low by the end of the experiment (Table 2). It is likely that the elevated phosphate levels, which were due to the addition of $H_3PO_4$ to reduce or maintain a stable pH, caused the iron to precipitate out of solution, making it unavailable to the plants. 'Rust' observed on the bottoms of both the 25% and 50% process water treatment containers supports this hypothesis.

**Table 2.** Results of analysis of the water solution (means) for the control (reverse osmosis: RO), 25% process water treatment (process $H_2O$ 25%), and 50% process water (process $H_2O$ 50%) treatments before and after the addition of Hoagland's solution (day 15), day 35, and day 44 in 2012. Values for undiluted process water (process $H_2O$ 100%) are shown for comparison. Note: TDN = total dissolved nitrogen; TDP = total dissolved phosphorus.

| | $NH_4^+$ (Nµg/L) | $NO_2^+$ $NO_3$ (Nµg/L) | TDN (Nµg/L) | TDP (Pµg/L) | Cl (mg/L) | $SO_4$ (mg/L) | Na (mg/L) | K (mg/L) | Ca (mg/L) | Mg (mg/L) | Fe (mg/L) | Al (mg/L) |
|---|---|---|---|---|---|---|---|---|---|---|---|---|
| Minimum level of detection | 2 | 1 | 10 | 3 | 0.03 | 0.04 | 0.016 | 0.009 | 0.005 | 0.01 | 0.016 | 0.004 |
| RO pre H * Day 15 | 27 | 289 | <LOD *** | <LOD | 5.51 | 31.03 | 13.05 | 0.90 | 31.75 | 7.61 | <LOD | <LOD |
| RO post H ** Day 15 | 428 | 3800 | <LOD | <LOD | 5.72 | 34.28 | 13.19 | 7.47 | 32.03 | 8.61 | 0.06 | <LOD |
| RO Day 35 | 17 | 42,867 | <LOD | 45,181 | 9.92 | 179.60 | 21.85 | 183.99 | 63.32 | 43.33 | 0.43 | <LOD |
| RO Day 44 | 43 | 37,933 | <LOD | 52,183 | 8.38 | 135.47 | 20.04 | 122.40 | 41.98 | 31.54 | 0.21 | <LOD |
| Process $H_2O$ 100% | 5787 | 16 | <LOD | <LOD | 717 | 397 | 1195 | 15 | 21 | 11 | <LOD | <LOD |
| Process $H_2O$ 25% pre H Day 15 | 2140 | 701 | <LOD | <LOD | 195.16 | 131.12 | 269.37 | 5.37 | 27.58 | 8.21 | <LOD | <LOD |
| Process $H_2O$ 25% post H Day 15 | 2390 | 2910 | <LOD | <LOD | 194.52 | 132.66 | 270.05 | 10.02 | 29.45 | 9.53 | 0.04 | <LOD |
| Process $H_2O$ 25% Day 35 | 10 | 50,567 | <LOD | 102,986 | 206 | 303 | 306 | 180 | 42 | 43 | 0.10 | 10 |
| Process $H_2O$ 25% Day 44 | 33 | 38,933 | <LOD | 109,417 | 198 | 257 | 275 | 120 | 30 | 32 | 0.06 | <LOD |
| Process $H_2O$ 50% pre H Day 15 | 4020 | 870 | <LOD | <LOD | 367.83 | 231.82 | 540.19 | 10.09 | 25.80 | 9.68 | <LOD | <LOD |
| Process $H_2O$ 25% post H Day 15 | 4060 | 2880 | <LOD | <LOD | 365.58 | 233.04 | 529.52 | 14.53 | 26.55 | 10.53 | 0.04 | <LOD |
| Process $H_2O$ 50% Day 35 | 26 | 33,933 | <LOD | 151,163 | 365.17 | 380.95 | 524.48 | 187.20 | 22.96 | 39.34 | 0.02 | <LOD |
| Process $H_2O$ 50% Day 44 | 27 | 53,467 | <LOD | 174,730 | 397.45 | 388.63 | 575.50 | 160.50 | 19.87 | 36.45 | 0.02 | <LOD |

* pre H = before addition of Hoagland's solution. ** post H = after addition of Hoagland's solution. *** LOD = limit of detection.

### 3.1.2. Clonal Variation

Fourteen clones consistently ranked in the top 30 (30 was selected as a target number of clones to ensure genetic diversity standards would be met [23]; Ne = 18 for operational deployment of native species from a CPP program onto public lands in Alberta) for stem height growth across all three treatments (Figure 1a). For stem basal diameter growth, 12 clones ranked consistently in the top 30 in all three treatments (Figure A1a). The same trends were observed for the top 12 clones in the 2013 experiment (Figure 1b; Figure A1b).

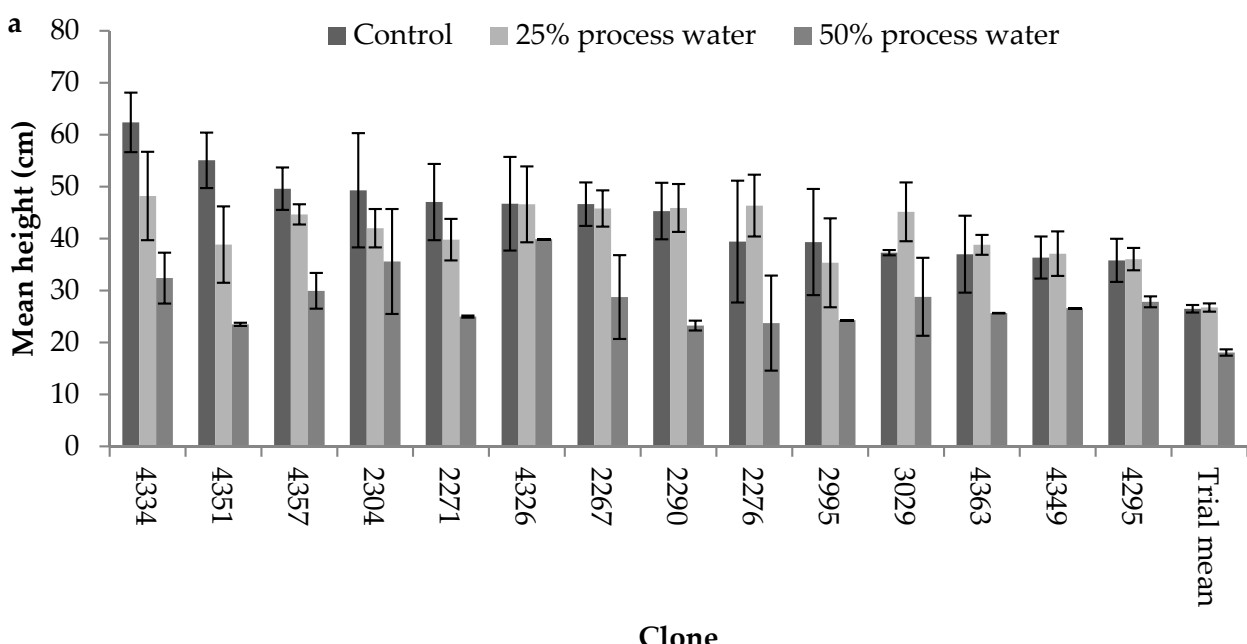

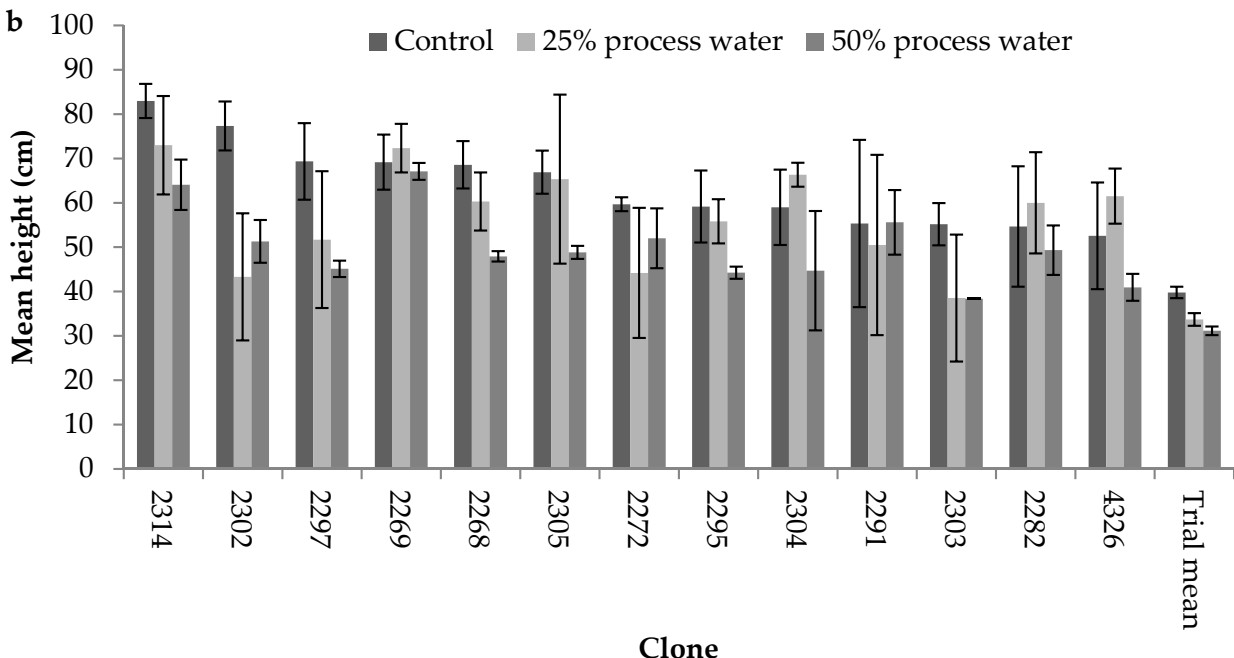

**Figure 1.** (**a**) Mean height (±SE) of 14 balsam poplar clones that ranked in the top 30 for all three water treatments as compared with the overall trial mean for 148 clones after 44 days of growth in the 2012 experiment. (**b**) Mean height (±SE) of 13 balsam poplar clones that ranked in the top 30 for all three water treatments as compared with the overall trial mean for 86 clones after 65 days of growth in the 2013 experiment.

Within the 148 clones tested in 2012, there were 38 clones that had higher total biomass in the 25% process water treatment than in the control treatment, while 24 clones had a higher mean total biomass in the 50% process water treatment than the control, and five clones ('AP2309', 'AP2453', 'AP3033', 'AP3127', and 'AP4356') showed the opposite trend with the 50% process-affected water treatment > 25% process-affected water treatment > control. Clones 'AP2453' and 'AP4356', which exhibited this reverse trend, also ranked within the top 30 clones for total biomass in all three treatments.

In the 2013 experiment, there was, again, a significant decrease in leaf, stem, root, and total biomass from the control to the 25%, and 50% treatment (Figure 2). Within the 86 clones that were tested in 2013, there were 27 clones that had higher total biomass for the 25% process water treatment than control (3.32 g ± 0.19), and 10 clones had higher total biomass in the 50% process water treatment than the control. See Appendix A Tables A1 and A2 for summary total biomass data for the 30 top performing clones in 2012 and 2013. Overall performance showed similar trends across both years for the top 10 clones (Figure 3). However, owing to the longer growth period in 2013, the biomass totals for 2013 were higher overall than those in 2012. There were, however, exceptions to this trend on an individual clone basis. In addition, some of the highest root/shoot ratios were observed in clones that had below average total biomass growth.

### 3.2. Field Testing
#### 3.2.1. Survival and Growth

Overall, all trees grew well at the edge of the end-pit lake. There was no significant difference between treatments for survival in 2019, which overall remained very high at 82%, 84%, and 85% for treatments 1, 2, and 3, respectively (Table 3). However, mortality rates increased from 11% (2015) to 17% (2019), which was likely due to higher mortality in the first row of trees adjacent to the lakeshore, with some having almost eroded into the lake.

**Table 3.** Per cent (%) survival rate (±SE) of balsam poplar among treatments (1 = 25 selected tolerant clones, 2 = 10 selected control clones, 3 = Stream 1 vegetative lot clones) in the fall of each year.

| Treatment | Year (Age) | | | | |
|:---:|:---:|:---:|:---:|:---:|:---:|
| | 2015 (Age 1) | 2016 (Age 2) | 2017 (Age 3) | 2018 (Age 4) | 2019 (Age 5) |
| 1 | 87% ± 2% | 87% ± 2% | 86% ± 2% | 83% ± 3% | 82% ± 3% |
| 2 | 92% ± 2% | 90% ± 3% | 90% ± 3% | 87% ± 4% | 84% ± 4% |
| 3 | 89% ± 4% | 89% ± 4% | 89% ± 4% | 87% ± 5% | 85% ± 5% |

The average height and DBH at year five for treatments 1, 2, and 3 were 3.75, 3.58, and 3.61 m for height (Figure 4) and 27.88, 27.02, and 26.38 mm for DBH, respectively (Figure 5). However, there were no significant differences in height and DBH among treatments. Mean growth increments for both height (Figure 4) and basal RCD or DBH (Figure 5) showed similar growth trends across all three treatments from 2015 to 2019. The largest annual height increment was in 2016, which averaged approximately 1 m for all three treatments (Figure 4).

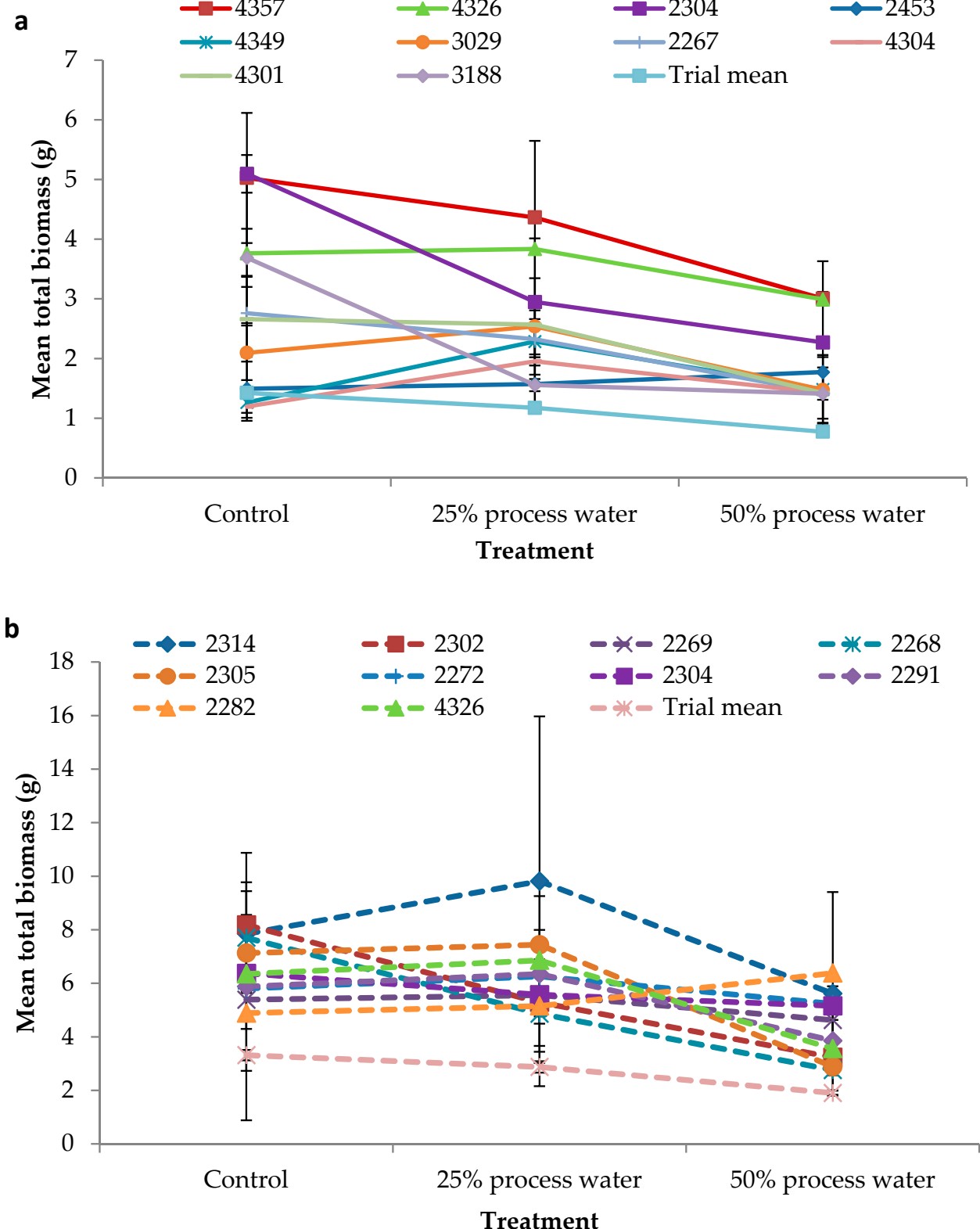

**Figure 2.** (**a**) Total biomass (mean dry weight $\pm$ SE) in each water treatment for the top 10 balsam poplar clones in the three water treatments and the overall means for all trees after 44 days of growth in the 2012 experiment (solid line). (**b**) Total biomass (mean dry weight $\pm$ SE) vs. treatment for the top 10 balsam poplar clones in the control, 25%, and 50% process water treatment solutions and the overall treatment mean after 65 days of growth in the 2013 experiment (dash line). Clones tested in both years (4326 and 2304) are in the same colour.

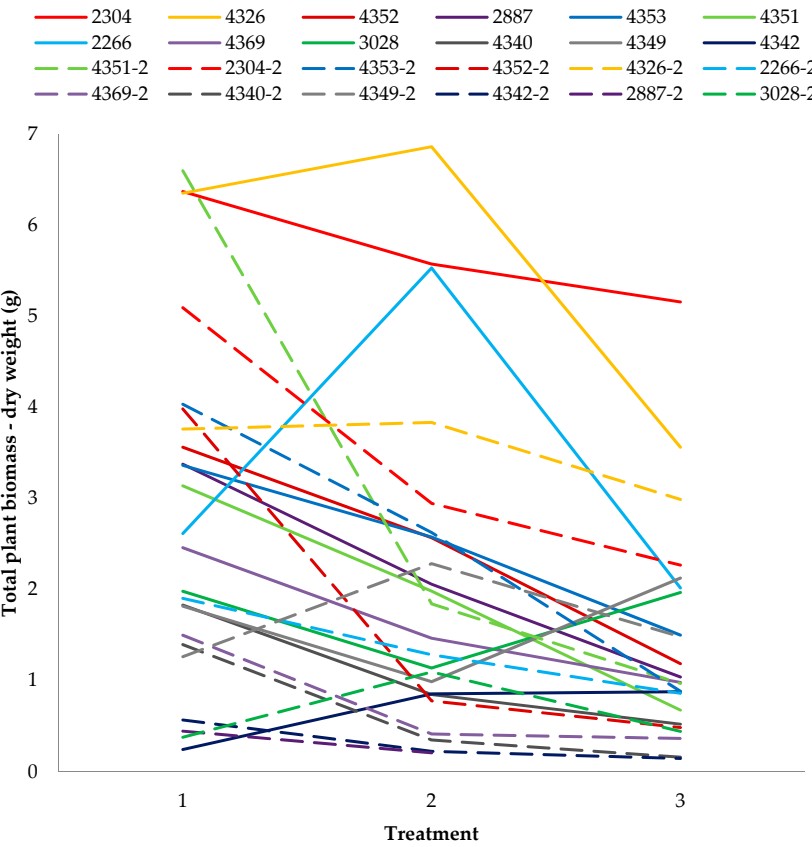

**Figure 3.** Performance (total plant biomass (g)) of the 12 balsam poplar clones grown in both 2012 and 2013 trials (2013 = solid line, 2012 = dashed line) under three treatments (1 = control; 2 = 25% process water; 3 = 50% process water).

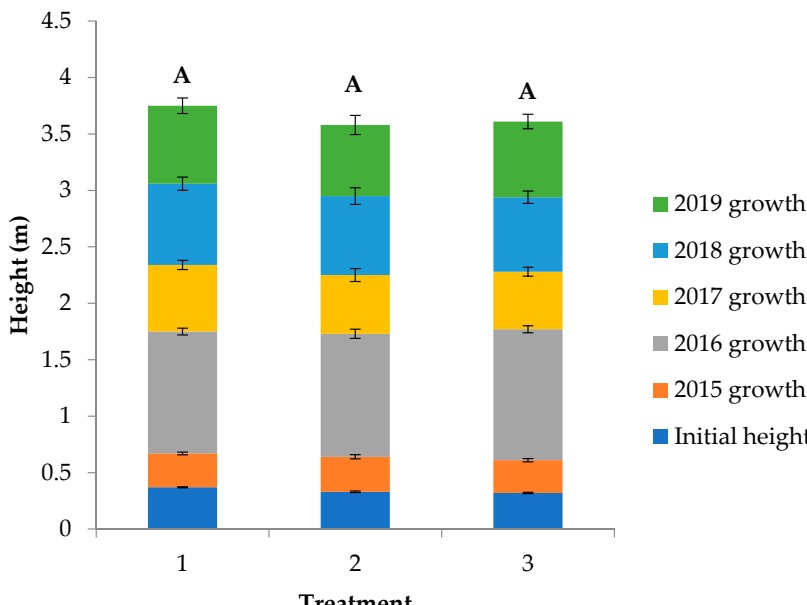

**Figure 4.** Mean initial height and mean annual height increment (±SE) (year 1 = 2015 growth; year 2 = 2016 growth; year 3 = 2017 growth; year 4 = 2018 growth; year 5 = 2019 growth) (m) (±SE) for balsam poplar trees planted in three treatments (1 = 25 selected tolerant clones, 2 = 10 selected control clones, 3 = Stream 1 veg. lot clones). Significant differences between treatment means for height are indicated by different letters at $p \leq 0.05$.

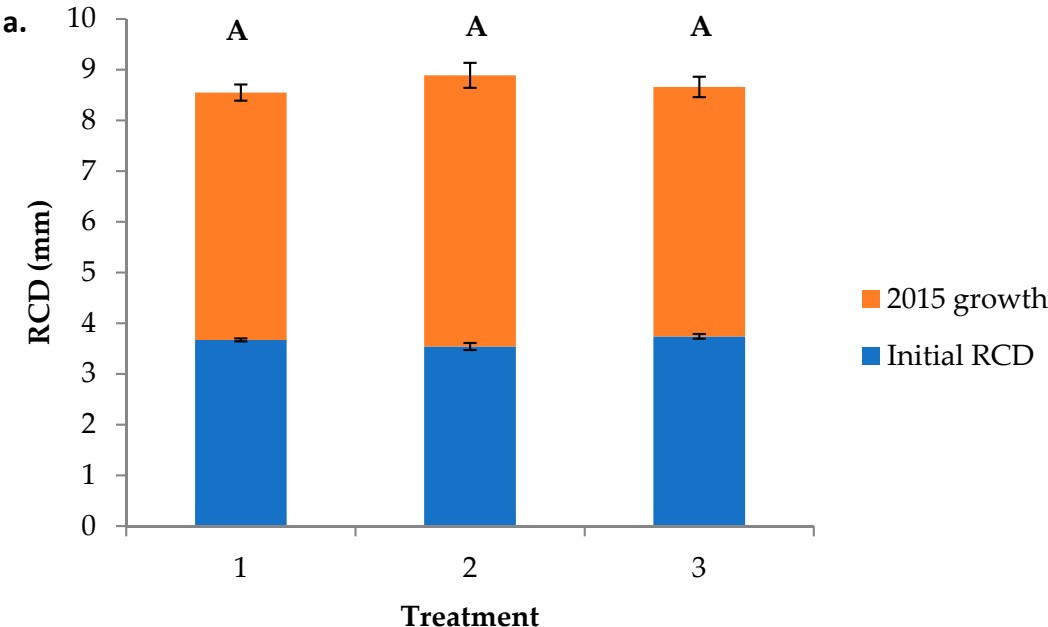

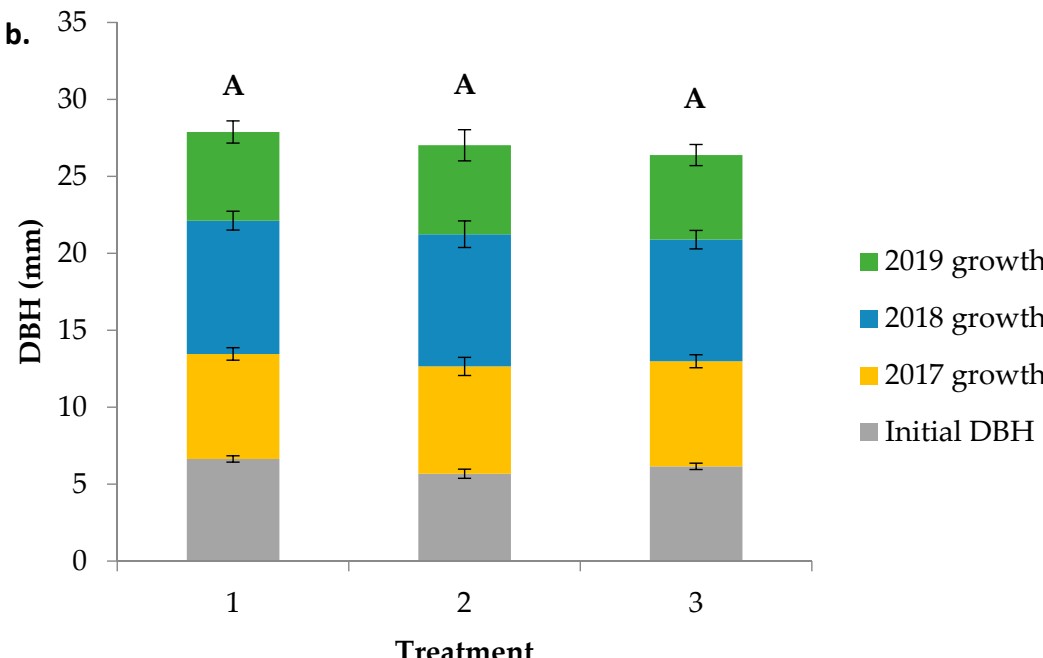

**Figure 5.** Mean initial diameter and mean annual growth increment (±SE) (year 1 = 2015 growth; year 2 = 2016 growth; year 3 = 2017 growth; year 4 = 2018 growth; year 5 = 2019 growth) (cm) (±SE) for balsam poplar trees planted in three treatments (1 = 25 selected tolerant clones, 2 = 10 selected control clones, 3 = Stream 1 veg. lot clones). (**a**) Basal root collar diameter (RCD mm) (2014–2015); (**b**) diameter at breast height (DBH mm) (2016–2019). Significant differences between treatment means for DBH are indicated by different letters at $p \leq 0.05$.

Significant differences were found among different blocks for both height and DBH growth parameters (Figure 6). The distance to the shoreline was used to determine the block design running parallel to the shore. Trees in Block 2 (10 m away to the lake edge) showed the best tree performance and this block represented the middle distance from the shoreline (between Block 1, closest to the water's edge, and Block 3, furthest up the slope from the water's edge).

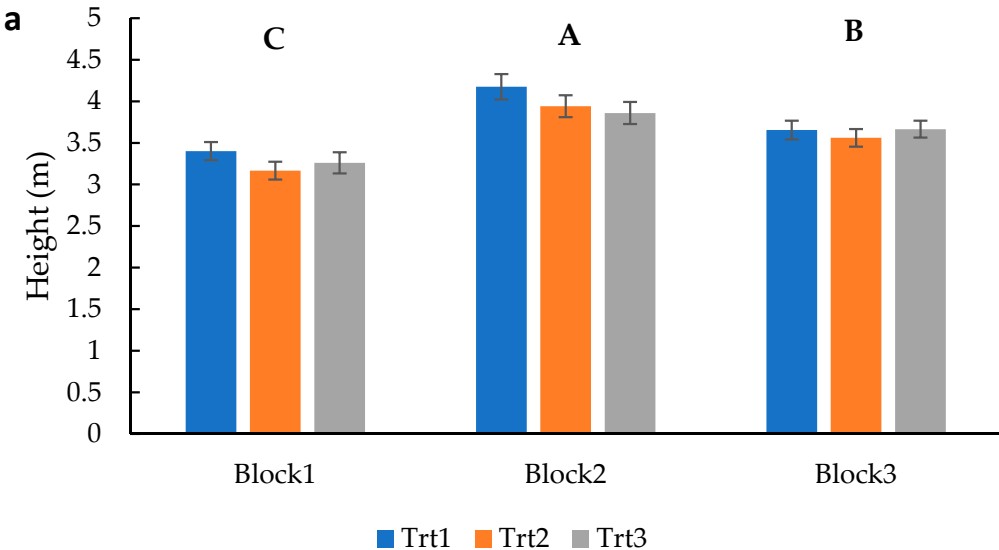

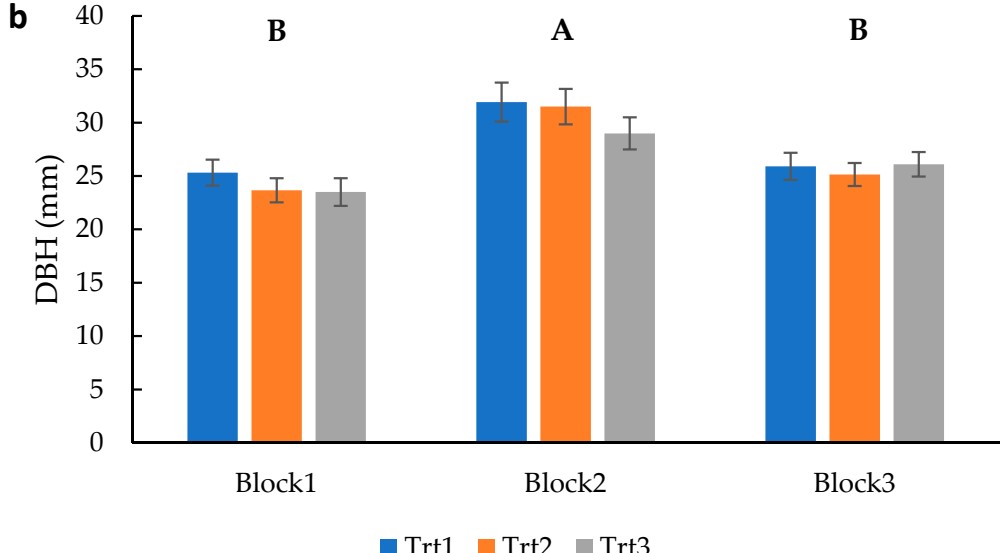

**Figure 6.** Treatment means (1 = 25 selected tolerant clones, 2 = 10 selected control clones, 3 = Stream 1 vegetative lot clones) of growth parameters for balsam poplar trees planted in three blocks (Block 1 = 5 m from the lake edge; Block 2 = 10 m from the lake edge; Block 3 = 15 m from the lake edge) in 2019. (**a**) Height (m); (**b**) DBH (mm). Significant differences between block means are indicated by different letters. Significant differences between treatment means for height or DBH in each block are indicated by different letters at $p \leq 0.05$.

Growth in height ranged from 2.5 m to more than 5 m across all clones (Figure 7a), while DBH ranged from 17 mm to more than 36 mm (Figure 7b). The Stream 1, treatment 3 clones showed average growth when compared with the 35 Stream 2 selected clones (treatment 1 + treatment 2) for both height and DBH (Figure 7a,b). Not surprisingly, there was a strong correlation (r = 0.925) between height and DBH by the fall of 2019, indicating that the taller trees also had, in general, great DBH.

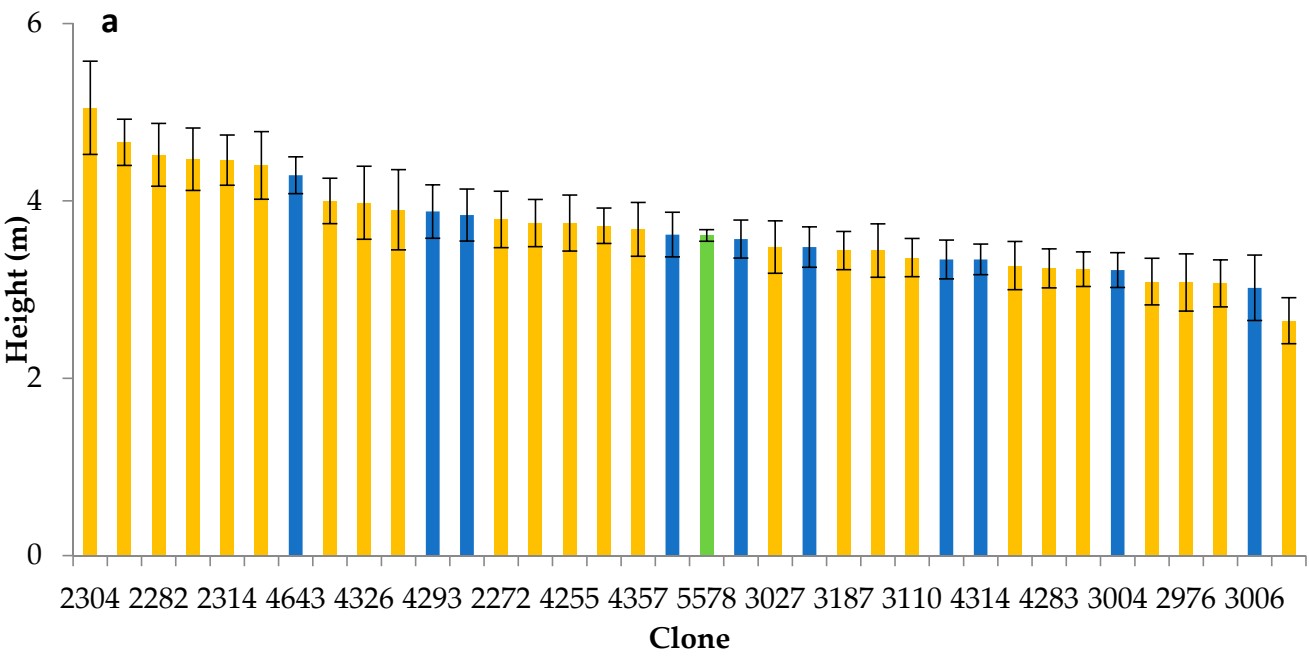

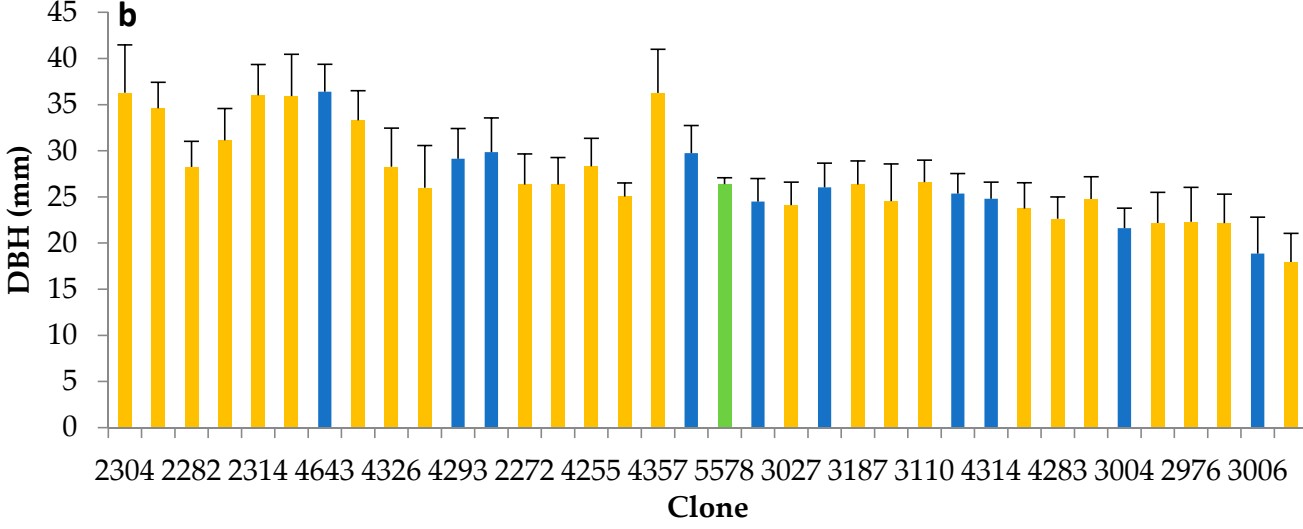

**Figure 7.** Mean growth (±SE) by balsam poplar clone (fall 2019). The yellow bars are treatment 1 clones, the dark blue bars are treatment 2 clones, and the green bar is treatment 3 (treatment 1 = 25 selected tolerant clones, treatment 2 = 10 selected control clones, treatment 3 = Stream 1 vegetative lot clones; # 5578 was the lot number for all Stream 1 clones). (**a**) Height (m); (**b**) DBH (mm).

When considering the tallest 18 clones from treatment 1 (where 25 clones were tested), which is the minimum number of clones required for unrestricted registration of a Stream 2 'lot' to be deployed operationally (i.e., Ne = 18) as determined by the government standards [23], the results showed a significant difference in height, but not DBH, when compared with treatment 2 (10 'low salt tolerant clones' from 50% process water testing) and, more importantly, the Stream 1 local 'wild' collection (treatment 3), for both height and DBH (Table 4).

**Table 4.** Growth data (height and diameter at breast height (DBH)) (±SE) of balsam poplar in 2019 for all treatments (top 18 clones from treatment 1 and all clones from treatment 2 and treatment 3). Significant differences between treatment means are indicated by different letters.

| Treatment/Lot Type | Height (m) | DBH (mm) |
|---|---|---|
| 1 (top 18 clones)/Stream 2 | 4.01 ± 0.08 [a] | 29.99 ± 0.86 [a] |
| 2 (10 control clones)/Stream 2 | 3.58 ± 0.09 [b] | 27.02 ± 1.00 [b] |
| 3 Local control/Stream 1 | 3.61 ± 0.06 [b] | 26.38 ± 0.69 [b] |

Stem volume was calculated for each tree using fall 2019 data based on the following equation: $V = A_b \times H/3$ (where V: stem volume ($cm^3$), $A_b$: basal area = $\pi \times DBH^2$ (diameter at breast height)/4 ($cm^2$), and H: height (cm)) [29] (Figure 8). Although no significant differences were found in either height or DBH, when stem volume was calculated, trees in treatment 1 (including all 25 clones) (1060.47 ± 68.24 $mm^3$) had a larger stem volume than trees in treatment 3 (814.87 ± 54.25 $mm^3$) (Figure 8). However, when considering the tallest 18 clones from treatment 1 (where 25 clones were tested), the stem volume of the top 18 clones was 1254.55 ± 86.71 $mm^3$, which is significantly greater than ($p < 0.05$) treatment 2 (893.39 ± 80.67 $mm^3$) and treatment 3.

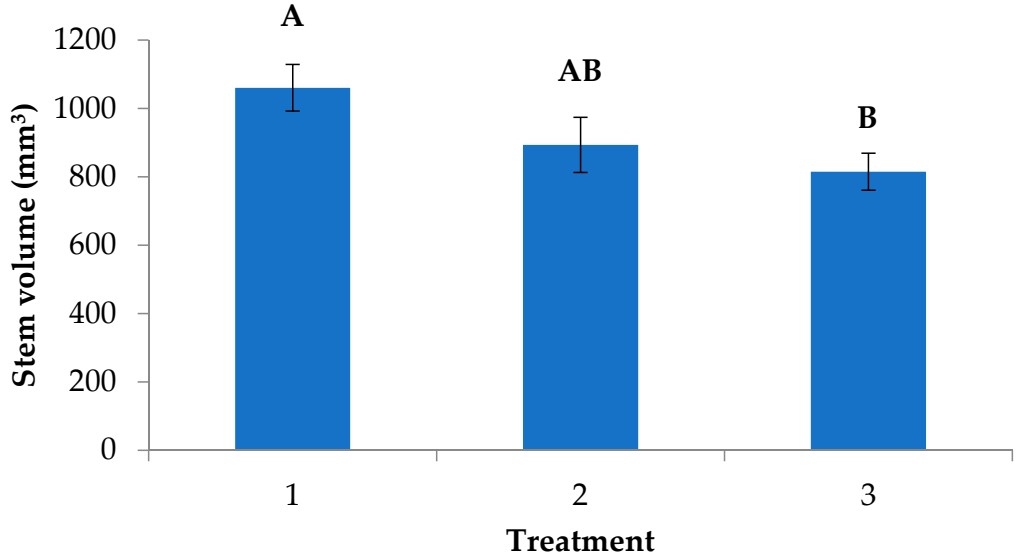

**Figure 8.** Mean stem volume ($mm^3$) (±SE) for five-year-old balsam poplar trees planted in 2019 (1 = 25 selected tolerant clones, 2 = 10 selected control clones, 3 = Stream 1 vegetative lot clones). Significant differences between treatment means for stem volume are indicated by different letters at $p \leq 0.05$.

### 3.2.2. Tissue Nutrient Analysis

Overall, nutrient analysis from bulk leaf tissue samples showed a low sodium level and a high calcium level (Table 5). There were no significant differences found for any of the nutrients by treatment except for magnesium, where treatment 3 had significantly higher levels compared with treatments 1 or 2 (Table 6). Significant differences were found among different blocks for some tissue nutrients (i.e., Cu, Fe, Mn, P, Zn, and N) for both treatments (Table 7).

**Table 5.** Mean tissue nutrient (a) and heavy metal (b) analysis (±SE) based on leaf tissue samples in two-year-old balsam poplar trees.

| a. Tissue nutrient | | | | | | | |
|---|---|---|---|---|---|---|---|
| **Site** | **B(ug/g)** | **Ca (%)** | **Cu (ug/g)** | **Fe (ug/g)** | **Mg (%)** | **Mn (ug/g)** | **Mo (ug/g)** |
| End-pit lake | 33.44 ± 0.82 | 1.05 ± 0.02 | 4.69 ± 0.12 | 136.06 ± 4.84 | 0.282 ± 0.004 | 9.65 ± 0.26 | 0.51 ± 0.01 |
| | **P (%)** | **K (%)** | **S (%)** | **Zn (ug/g)** | **N (%)** | **Na (%) *** | |
| | 0.164 ± 0.003 | 1.45 ± 0.03 | 0.36 ± 0.01 | 127.48 ± 5.51 | 1.38 ± 0.06 | - | |
| b. Heavy metal ** | | | | | | | |
| **Site ***** | **Al (ug/g)** | **Ba (ug/g)** | **Cd (ug/g)** | **Cr (ug/g)** | **Co (ug/g)** | **Li (ug/g)** | **Ni (ug/g)** |
| End-pit lake | 55.90 ± 14.70 | 1.87 ± 0.27 | 0.20 ± 0.00 | 0.79 ± 0.26 | 0.38 ± 0.13 | 0.45 ± 0.05 | 0.74 ± 0.11 |
| | **Si (ug/g)** | **Sr (ug/g)** | **Sn (ug/g)** | **Ti (ug/g)** | **V (ug/g)** | | |
| | 270.67 ± 47.10 | 11.73 ± 1.53 | 1.40 ± 0.06 | 2.30 ± 0.30 | 3.83 ± 0.12 | | |

* Na levels were all <0.01, so no statistical analysis could be completed. ** Heavy metal analysis was only conducted on Stream 1 control trees. *** Metal levels <0.5 (ug/g) are not shown.

**Table 6.** Mean leaf tissue nutrient analysis (±SE) results by treatment in two-year-old balsam poplar trees. Significant differences between treatment means at $p \leq 0.05$ are indicated by different letters. Treatment 1 = 25 selected tolerant clones, treatment 2 = 10 selected control clones, treatment 3 = Stream 1 vegetative lot clones.

| Treatment | B(ug/g) | Ca (%) | Cu (ug/g) | Fe (ug/g) | Mg (%) | Mn (ug/g) | Mo (ug/g) |
|---|---|---|---|---|---|---|---|
| 1 | 33.62 ± 1.02 | 1.04 ± 0.02 | 4.42 ± 0.13 | 139.83 ± 6.47 | 0.283 ± 0.004 [ab] | 9.56 ± 0.33 | 0.48 ± 0.04 |
| 2 | 32.88 ± 1.52 | 1.07 ± 0.04 | 5.39 ± 0.21 | 127.00 ± 6.16 | 0.275 ± 0.007 [b] | 9.67 ± 0.44 | 0.55 ± 0.04 |
| 3 | 33.70 ± 0.55 | 0.97 ± 0.05 | 4.61 ± 0.57 | 132.33 ± 14.44 | 0.32 ± 0.026 [a] | 8.70 ± 1.79 | 0.47 ± 0.03 |

| Treatment | P (%) | K (%) | S (%) | Zn (ug/g) | N (%) | Na (%) | |
|---|---|---|---|---|---|---|---|
| 1 | 0.161 ± 0.003 | 1.46 ± 0.03 | 0.35 ± 0.01 | 119.06 ± 6.61 | 1.35 ± 0.07 | - | |
| 2 | 0.172 ± 0.006 | 1.44 ± 0.04 | 0.37 ± 0.01 | 149.42 ± 9.48 | 1.43 ± 0.13 | - | |
| 3 | 0.160 ± 0.011 | 1.46 ± 0.12 | 0.33 ± 0.04 | 118.47 ± 10.90 | 1.38 ± 0.31 | - | |

* Na levels were all <0.01, so no statistical analysis could be completed.

**Table 7.** Block means of tissue nutrient analysis for each treatment in two-year-old balsam poplar trees. Significant differences between block means at $p \leq 0.05$ are indicated by different letters. Block 1 = 5 m from the lake edge; Block 2 = 10 m from the lake edge; Block 3 = 15 m from the lake edge.

| a. Treatment 1 (25 selected tolerant clones) | | | | | | | |
|---|---|---|---|---|---|---|---|
| **Block** | **B (ug/g)** | **Ca (%)** | **Cu (ug/g)** | **Fe (ug/g)** | **Mg (%)** | **Mn (ug/g)** | **Mo (ug/g)** |
| 1 | 34.14 ± 1.74 | 1.06 ± 0.04 | 5.19 ± 0.20 [a] | 178.44 ± 6.29 [a] | 0.270 ± 0.008 | 10.96 ± 0.69 [a] | 0.58 ± 0.03 |
| 2 | 33.65 ± 1.50 | 1.03 ± 0.05 | 4.46 ± 0.19 [b] | 123.68 ± 6.58 [b] | 0.280 ± 0.008 | 8.94 ± 0.43 [b] | 0.53 ± 0.04 |
| 3 | 33.16 ± 2.07 | 1.02 ± 0.04 | 3.65 ± 0.18 [c] | 115.36 ± 7.47 [c] | 0.300 ± 0.008 | 8.77 ± 0.47 [b] | 0.42 ± 0.04 |
| **Block** | **P (%)** | **K (%)** | **S (%)** | **Zn (ug/g)** | **N (%)** | **Na (%)** | |
| 1 | 0.170 ± 0.007 [a] | 1.55 ± 0.04 | 0.36 ± 0.01 | 137.54 ± 9.69 [a] | 1.81 ± 0.07 [a] | - | |
| 2 | 0.170 ± 0.007 [a] | 1.44 ± 0.07 | 0.36 ± 0.02 | 115.71 ± 8.26 [b] | 1.34 ± 0.09 [b] | - | |
| 3 | 0.140 ± 0.005 [b] | 1.41 ± 0.07 | 0.33 ± 0.02 | 100.94 ± 7.71 [c] | 0.89 ± 0.14 [c] | - | |
| b. Treatment 2 (10 selected control clones) | | | | | | | |
| **Block** | **B (ug/g)** | **Ca (%)** | **Cu (ug/g)** | **Fe (ug/g)** | **Mg (%)** | **Mn (ug/g)** | **Mo (ug/g)** |
| 1 | 35.65 ± 2.70 [a] | 1.09 ± 0.06 | 6.22 ± 0.38 [a] | 166.80 ± 4.50 [a] | 0.290 ± 0.010 | 10.98 ± 0.92 [a] | 0.62 ± 0.02 |
| 2 | 32.39 ± 2.29 [b] | 1.08 ± 0.09 | 5.10 ± 0.28 [b] | 112.40 ± 5.62 [b] | 0.270 ± 0.020 | 9.36 ± 0.74 [a] | 0.53 ± 0.03 |
| 3 | 30.60 ± 2.91 [b] | 1.03 ± 0.05 | 4.84 ± 0.33 [b] | 101.80 ± 6.70 [b] | 0.270 ± 0.009 | 8.67 ± 0.44 [b] | 0.57 ± 0.03 |
| **Block** | **P (%)** | **K (%)** | **S (%)** | **Zn (ug/g)** | **N (%)** | **Na (%)** | |
| 1 | 0.180 ± 0.012 [a] | 1.48 ± 0.07 | 0.39 ± 0.02 | 183.00 ± 8.17 [a] | 1.77 ± 0.25 [a] | - | |
| 2 | 0.172 ± 0.012 [a] | 1.47 ± 0.06 | 0.39 ± 0.02 | 152.00 ± 9.23 [b] | 1.56 ± 0.15 [b] | - | |
| 3 | 0.160 ± 0.007 [b] | 1.36 ± 0.09 | 0.34 ± 0.03 | 113.30 ± 11.01 [c] | 0.95 ± 0.17 [c] | - | |

## 4. Discussion

### 4.1. Greenhouse Study

Interest in and acceptance of poplar and willow for use in reclamation and phytoremediation have been increasing in the last 15 years [30,31]. Salinity is known to reduce water absorption and cause water stress [32]. Salts are taken up by plants and the increasing tissue ion concentration contributes to a decrease in water potentials. In addition, some plant species accumulate solutes under stress to maintain a positive water balance [33]. This osmotic adjustment allows plants to maintain turgor in saline environments. Approximately 1% of all plant species are halophytes and can complete their life cycle in relatively high saline environments, such as 200 mM NaCl or more [34]. The ability of poplars to grow on harsh sites, along with being relatively easy to propagate, through the use of cuttings, makes them an ideal candidate species for use in reclamation, more specifically with respect to this study, on reclaimed oil sands mine sites in northeastern Alberta.

In both years of the greenhouse studies, clear differences in the phenotypic growth response of the tested balsam poplar clones were observed, indicating tolerance for OSPW by native balsam poplars and clonal variability in that tolerance. These findings support the assertion that the opportunity exists to select and propagate an easily propagated native species for use in reclaiming these challenging sites. More specifically, there was a high degree of genetic (clonal) variability in survival, height, diameter, and biomass growth in response to the control, 25%, and 50% process water treatments. Most clones performed more poorly in the 25% and 50% process water solutions as compared with the control. There were several clones, however, that performed consistently better than the average for all of the traits measured across all three treatments, exhibiting desirable traits for selection from the population of clones tested. These results suggest that genetic differences in clones should be considered in the selection of genetic materials for use on reclamation sites impacted by high salt-containing tailings generated from oil sands operations.

Clone 'AP4357', tested in both years of the study, is an example of an OSPW-tolerant clone that consistently performed at or near the top for all traits measured, and in all three water treatments across both years. There were also a number of clones that performed better in the 25% process water treatment, 50% process water treatment, or both as compared with their control treatment performance. These tolerant clones appeared to actually prefer the saline conditions, which indicates that there are balsam poplar clones that are salt loving or 'halophiles'.

Significant positive correlations between height growth and rooting traits such as root length and root dry weight of poplar have been reported [35]. Thus, we believe height growth can be used as a surrogate measure of root development, which may have increased associated microbial activity in the rhizosphere. Therefore, our better-rooting clones (i.e., clones 'AP4326' and 'AP2304') may exhibit greater remedial potential. In addition, root/shoot ratios are often very useful in determining if plants have healthy root systems relative to above ground biomass [36]. Because salty soils often limit root penetration and inhibit root growth [37], clones that have higher root growth are likely going to have increased performance in saline environments. Therefore, higher root/shoot ratios would be desirable. However, one must be careful not to look at root/shoot ratios as a single trait for selection as it gives no indication of the actual growth performance of the plant. Ideally, in the selection of suitable clones from this experiment, clones that have high total above ground biomass with an above average root/shoot ratio would be considered desirable.

### 4.2. Field Testing

#### 4.2.1. Survival and Growth

The initial high survival rate of all treatments at the end-pit lake (Table 3) indicated these trees were well adapted to the reclamation mine site. Additionally, survival through the first two years was high, indicating that early survival is an important indicator of later survival and growth. Owing to good water availability and sufficient nutrients throughout

each growing season, trees grew very well (Figures 4 and 5). The trees also experienced little to no competition owing to the installed black plastic, which also likely made the soil warmer, although this was not measured. In this trial, distance to the shoreline had a significant impact on the trees' height and DBH (Figure 6). For trees planted adjacent to the shoreline, the trees had ready access to the water table while also being more vulnerable to shoreline erosion.

As no significant differences were found between treatment 3 (Stream 1 vegetative lot clones) and the other two plant treatment groups for height and DBH, these results suggested the growth rates of the Stream 1 vegetative lot clones might be considered acceptable when compared with the Al-Pac selected clones (salt-tolerant and controls from Al-Pac's program) on a site with 'ideal' growing conditions. However, mean stem volume showed a significant difference between treatments (treatment 1 ≥ treatment 2 ≥ treatment 3), indicating the selected Stream 2 clones from Al-Pac's CPP program, overall, grew better than the Stream 1 vegetative lot clones. In addition, analysis of the top 18 clones from the Stream 2 lot selected from the treatment 1 group showed significant differences for both height and DBH when compared with treatments 2 and 3 (Table 4). This suggests that there is potential to plant groups of selected clones that would outperform wild clonal collections of Stream 1 native balsam, with the added advantage of not being restricted to deployment only within their local seed zone, but taking advantage of the entire region associated with the balsam poplar controlled parentage program.

Interestingly, despite clones selected from treatment 2 being chosen from the group of clones that did not exhibit superior salt tolerance (i.e., greatest growth) in the initial greenhouse screening trials (2012 and 2013), they still showed a level of tolerance to salinity testing [22]. It is worth noting that the selected high tolerance clones (treatment 1) were not necessarily the tallest trees overall in the initial screening experiment; some of the clones performed well in all three treatments, while others were selected because they performed better in the 50% process water treatment than in the 25% and control treatments. Moreover, the EC level of the 50% process water in the greenhouse was between 3 and 3.6 mS cm$^{-1}$ [22]; however, the EC level of surface water from the end-pit lake was only 2.7–3.0 mS cm$^{-1}$ (salty water) [37], which suggests that the Al-Pac selected clones did not experience the same level of stress in the field as their previous screening showed they had the capability to sustain. Therefore, if the end-pit lake site was not, in fact, heavily impacted by salts, the selected high salt tolerant treatment clones that did the best in the greenhouse trial might not have had the opportunity to exhibit their superior 'salt tolerance', and thus to this point in time, looked similar to the clones represented in treatment 2 and in the Stream 1 vegetative lot.

From the current data, Stream 2 clones, selected for salt tolerance, had greater volume when compared with the Stream 1 wild lot, even though for height and DBH alone, clones from treatment 1 and treatment 2 showed growth increments that were interspersed with each other, and treatment 3 was close to the median in performance (Figure 7). As treatment 1 (Stream 2) clones showed salt tolerance in the greenhouse study and performed well in the field study, selecting and planting these trees may prove beneficial in the future if reclamation sites become more challenging. In addition, access to the Stream 2 clones from stoolbeds and/or existing trees could simplify collections while also ensuring the material can be planted over a much wider area (i.e., no seed zone restrictions) associated with the CPP.

### 4.2.2. Nutrient Analysis

Sodium and calcium were of particular interest in this experiment as they are the main drivers of potential 'salinity' conditions on reclamation sites [38]. White and Liber (2018) [37] characterized the chemical constituents in surface water from this end-pit lake, and found sodium was one of the main ions that contributed to salinity. However, foliar sodium levels in our current study were below any accurately detectable level and, as such, were reported as being <0.01% (Table 5a). Foliar calcium levels, which were in the

normal value range (from 0.1% to 5%) [39], however, were higher than sodium levels in the samples (Table 5a). The low sodium levels measured may suggest that sodicity is currently not a concern, or that the poplars did not accumulate it in their leaves. The heavy metal accumulation patterns were similar to the published results under field conditions [40,41] (Table 5b), where Zn and Cd were accumulated in the leaves, indicating the phytoextraction ability of balsam poplar. The significant differences found in magnesium (Mg) levels among the three treatments indicate that the Stream 1 vegetative lot clones could be potentially used in a high Mg contaminated field. Overall, the trend from tissue nutrient analysis using block means (Table 7) showed the nutrient levels were higher in the block closest to the water. This finding indicated that a high-water table might also offer the opportunity for salt accumulation in both the immediately adjacent surrounding soil and eventually in the plant tissues that could affect tree growth. In addition, the data from the heavy metal analysis could be used as supplementary data that might be useful in the future as a source for comparison. However, there is very limited published literature that outlines the range of acceptable nutrient concentrations in balsam poplar leaves.

## 5. Conclusions

In consideration of the results obtained from both the greenhouse and field studies, there is an opportunity to select genetically suitable native clones of balsam poplar that are tolerant to challenging growing conditions, making them more suitable for planting in reclamation efforts on potentially saline sites than unselected clones or populations. The field testing indicated the potential use of selected Stream 2 clones (selected high salt-tolerant clones) from Al-Pac's balsam poplar controlled parentage program for oil sands reclamation sites in northeastern Alberta. In addition, the Stream 1 wild lot showed comparable growth performance under "ideal" conditions. However, if reclamation were being conducted on challenging, salty sites, clones from the Al-Pac selections are recommended. Furthermore, the selected salt-tolerant clones showed greater stem volumes, which indicates that they are potentially the most flexible trees as they will likely do better under higher salt conditions, and they will have a greater volume even when conditions are favourable. Balsam poplar has shown considerable genetic diversity in growth performance in this study and such results are encouraging in light of an expanding industrial energy sector footprint. Moreover, poplars are well known for their ability to tolerate salinity [17,18,42] and, therefore, screening clones for salt tolerance, and maximizing the potential use of the tree improvement program trees available through Al-Pac, while meeting government regulations for genetic diversity, could provide a significant opportunity for reclamation in the oil sands region in Alberta. Reclamation challenges are in their infancy in Alberta and adjacent regions, and selected material from native species may provide greater benefit as a source of reclamation materials than untested material to help meet those challenges.

**Author Contributions:** B.R.T. and D.K. conceived and designed the experiments. Y.H., D.K., and B.R.T. performed the experiments and analyzed the data. Y.H., B.R.T., and D.K. wrote the paper. R.K. contributed nutrient solutions and the aeroponic system in the greenhouse study. Y.H., D.K., R.K., and B.R.T. reviewed and edited the paper. All authors have read and agreed to the published version of the manuscript.

**Funding:** This research was funded by Syncrude Canada Ltd. through a contract with Alberta-Pacific Forest Industries Inc. (Al-Pac). In addition, Y.H. received funding through MITACS and Al-Pac.

**Data Availability Statement:** The data presented in this study are available on request from the corresponding author.

**Acknowledgments:** This experiment has been supported by Alberta-Pacific Forest Industries Inc. and received considerable logistic assistance from Syncrude Canada Ltd. We thank the MITACS program and Al-Pac for providing support to Y.H. to complete this project. Thanks, is also given to Nathalie Startsev, Chen Ding, and Marc Robbins for technical assistance in the greenhouse experiments. The authors wish to express their gratitude to Craig Farnden (Syncrude Canada Ltd.) for his assistance in field data collection and logistics. The authors also thank Ellen Macdonald (University of Alberta) for her valuable suggestions and assistance during the greenhouse phase of the project.

**Conflicts of Interest:** The authors declare no conflict of interest.

## Appendix A

**Table A1.** Mean total biomass (±SE, grams) of the top 30 balsam poplar clones in the 50% process water treatment with corresponding means (±SE) for the control and 25% process water treatments in 2012. Clone numbers in bold identify clones that ranked in the top 30 for all three water treatments, while underlined clone numbers indicated consistent performance in both years' experiments. Periods indicate dead plants.

| | Control | | 25% Process Water | | 50% Process Water | |
|---|---|---|---|---|---|---|
| **Clone** | **Mean (g)** | **±SE** | **Mean (g)** | **±SE** | **Mean (g)** | **±SE** |
| **4357** | 5.03 | 0.63 | 4.36 | 0.26 | 3.00 | 0.63 |
| **<u>4326</u>** | 3.76 | 1.65 | 3.84 | 0.25 | 2.99 | 0.85 |
| **<u>2304</u>** | 5.10 | 0.32 | 2.95 | 1.07 | 2.27 | 1.36 |
| 2453 | 1.49 | 0.05 | 1.57 | 1.11 | 1.77 | 0.45 |
| <u>4349</u> | 1.27 | 0.07 | 2.29 | | 1.48 | 0.68 |
| **3029** | 2.10 | 0.54 | 2.54 | 0.36 | 1.47 | |
| **2267** | 2.76 | 0.45 | 2.32 | | 1.43 | |
| 4304 | 1.20 | 1.96 | 1.96 | 0.28 | 1.43 | |
| **4301** | 2.66 | 0.46 | 2.57 | 0.81 | 1.42 | 0.55 |
| 3188 | 3.69 | 0.21 | 1.56 | 0.76 | 1.41 | 0.12 |
| **2995** | 3.26 | 0.47 | 2.60 | 0.40 | 1.39 | |
| **4363** | 2.47 | 0.15 | 2.02 | 0.11 | 1.38 | |
| 4255 | 1.92 | 0.49 | 1.02 | 0.10 | 1.36 | 0.10 |
| 4296 | 3.93 | 0.18 | 1.07 | 0.01 | 1.36 | 0.08 |
| **4334** | 5.62 | 0.15 | 2.58 | 0.47 | 1.35 | 0.07 |
| 4277 | 1.16 | 0.34 | 2.57 | 0.48 | 1.34 | 0.25 |
| 4295 | 1.63 | 0.29 | 1.26 | 0.73 | 1.32 | 0.61 |
| **2288** | 4.01 | 0.27 | 1.86 | 0.28 | 1.26 | 0.08 |
| 3110 | 1.70 | 1.24 | 0.99 | 0.09 | 1.22 | 0.08 |
| 4285 | 1.53 | 0.22 | 4.17 | | 1.19 | |
| 4315 | 0.23 | 0.71 | 2.15 | 0.24 | 1.19 | 0.43 |
| 3187 | 0.70 | 0.19 | 0.62 | 0.71 | 1.14 | . |
| 2447 | 1.63 | 0.03 | 0.24 | 0.25 | 1.13 | 0.17 |
| 4249 | 0.91 | 0.21 | 1.61 | 0.32 | 1.11 | 0.30 |
| 2976 | 1.26 | 1.18 | 0.54 | 1.81 | 1.09 | |
| 2312 | 0.41 | 0.97 | 2.47 | 0.91 | 1.08 | 0.31 |
| 4317 | 1.56 | 0.17 | 1.23 | 0.70 | 1.06 | |
| 4297 | 0.61 | | 0.36 | | 1.02 | 0.06 |
| 4356 | 0.66 | 1.09 | 0.89 | 0.02 | 1.02 | 0.12 |
| 3106 | 1.02 | 0.91 | 2.30 | 0.74 | 1.01 | 0.13 |
| **Treatment mean** | 1.42 | 0.08 | 1.17 | 0.06 | 0.77 | 0.04 |

**Table A2.** Mean total biomass (±SE, grams) of the top 30 balsam poplar clones in the 50% process water treatment with corresponding means (±SE) for the control and 25% process water treatments in 2013. Clone numbers in bold identify clones that ranked in the top 30 for all three water treatments, while underlined clone numbers indicated consistent performance in both years' experiments. Periods indicate dead plants.

| | Control | | 25% Process Water | | 50% Process Water | |
|---|---|---|---|---|---|---|
| **Clone** | **Mean (g)** | **±SE** | **Mean (g)** | **±SE** | **Mean (g)** | **±SE** |
| **2282** | 4.89 | 2.16 | 5.16 | 0.40 | 6.39 | 3.02 |
| **2314** | 7.85 | 0.56 | 9.82 | 6.16 | 5.62 | 0.27 |
| **2287** | 8.91 | 5.21 | 9.85 | 3.71 | 5.55 | 0.68 |
| **2272** | 5.80 | 0.39 | 6.26 | 0.98 | 5.24 | 0.87 |
| **2304** | 6.37 | 1.65 | 5.57 | 1.39 | 5.16 | 2.02 |
| **2313** | 3.92 | 1.21 | 4.81 | 2.03 | 5.01 | 0.24 |
| **2269** | 5.39 | 1.09 | 5.56 | 1.07 | 4.63 | 0.88 |
| **2291** | 5.88 | 4.99 | 6.35 | 2.91 | 3.86 | 0.98 |
| **4326** | 6.35 | 3.52 | 6.86 | 1.03 | 3.56 | 0.37 |
| 2301 | 3.40 | 1.95 | 2.26 | 1.63 | 3.43 | 0.72 |
| **2302** | 8.18 | 1.26 | 5.29 | 2.27 | 3.24 | 1.40 |
| **2289** | 3.67 | 1.89 | 2.97 | 1.13 | 3.14 | 1.34 |
| **2305** | 7.13 | 1.42 | 7.44 | 0.55 | 2.89 | 0.32 |
| **2300** | 6.42 | 1.91 | 5.61 | 1.34 | 2.88 | 0.47 |
| **2268** | 7.70 | 2.07 | 4.86 | 2.71 | 2.76 | 0.77 |
| 2278 | 4.85 | 1.46 | 2.81 | 1.15 | 2.63 | 0.77 |
| 3027 | 5.33 | 1.17 | 1.46 | 0.013 | 2.59 | 0.30 |
| **2307** | 4.71 | 2.63 | 3.86 | 1.47 | 2.58 | 0.64 |
| 2997 | 2.68 | 1.90 | 1.26 | 0.72 | 2.47 | 0.44 |
| 2284 | | | 1.985 | | 2.3 | |
| **2297** | 8.50 | 1.93 | 8.71 | 0.64 | 2.22 | 0.14 |
| **2295** | 4.20 | 1.55 | 3.27 | 0.35 | 2.21 | 0.32 |
| 4274 | 1.05 | 0.43 | 1.55 | 0.12 | 2.16 | |
| **4349** | 1.82 | 1.23 | 0.99 | 0.64 | 2.13 | 0.28 |
| 4283 | 3.05 | 0.64 | 0.88 | 0.51 | 2.08 | 0.41 |
| 2266 | 2.62 | 0.27 | 5.53 | 2.93 | 2.02 | 0.68 |
| **2303** | 4.93 | 0.75 | 4.16 | 0.90 | 1.99 | |
| 2293 | 3.61 | 0.47 | 3.52 | 0.87 | 1.99 | 0.26 |
| 915 | 4.27 | 0.86 | 2.70 | 1.63 | 1.97 | 0.30 |
| 3028 | 1.98 | 0.24 | 1.14 | 0.048 | 1.97 | 0.68 |
| **Treatment mean** | 3.32 | 0.19 | 2.88 | 0.21 | 1.90 | 0.10 |

**Table A3.** Growth variables of balsam poplar tested by ANOVA indicating the source of variation, F-value, degrees of freedom (df), and *p*-value for the 2012 and 2013 greenhouse experiments for different clones, treatments, and interaction effects. Trt = treatment.

| | | **Growth Season** | | | | | |
|---|---|---|---|---|---|---|---|
| | | *2012* | | | *2013* | | |
| **Growth variable** | **Source of variation** | F | df | *p*-value | F | df | *p*-value |
| Stem height (cm) | Clone | 5.26 | 144 | <0.001 | 9.24 | 85 | <0.001 |
| | Trt | 85.78 | 2 | <0.001 | 31.64 | 2 | <0.001 |
| | Clone * Trt | 1.17 | 237 | 0.091 | 0.83 | 169 | 0.929 |
| Basal diameter (mm) | Clone | 3.83 | 144 | <0.001 | 6.24 | 85 | <0.001 |
| | Trt | 7.86 | 2 | 0.004 | 23.28 | 2 | <0.001 |
| | Clone * Trt | 1.01 | 237 | 0.471 | 0.92 | 169 | 0.730 |
| Stem biomass (g) | Clone | 4.65 | 144 | <0.001 | 6.36 | 85 | <0.001 |
| | Trt | 51.64 | 2 | <0.001 | 35.82 | 2 | <0.001 |
| | Clone * Trt | 1.45 | 237 | 0.007 | 0.84 | 169 | 0.897 |
| Root biomass (g) | Clone | 4.6 | 144 | <0.001 | 4.29 | 85 | <0.001 |
| | Trt | 39.32 | 2 | <0.001 | 25.4 | 2 | <0.001 |
| | Clone * Trt | 1.23 | 237 | 0.039 | 0.75 | 169 | 0.985 |
| Leaf biomass (g) | Clone | 4.51 | 144 | <0.001 | 5.62 | 85 | <0.001 |
| | Trt | 39.56 | 2 | <0.001 | 30.31 | 2 | <0.001 |
| | Clone * Trt | 1.17 | 237 | 0.082 | 0.99 | 169 | 0.538 |
| Total biomass (g) | Clone | 4.97 | 144 | <0.001 | 5.77 | 85 | <0.001 |
| | Trt | 47.06 | 2 | <0.001 | 34.60 | 2 | <0.001 |
| | Clone * Trt | 1.28 | 237 | 0.016 | 0.87 | 169 | 0.841 |

**Table A4.** ANOVA for mean stem volume ($mm^3$) of balsam poplar in 2019 at the end-pit lake (treatment 1 = 25 selected tolerant clones, treatment 2 = 10 selected control clones, treatment 3 = Stream 1 vegetative lot clones).

| **Source** | **DF** | **F Value** | ***p*-Value** |
|---|---|---|---|
| Block | 2 | 19.95 | <0.0001 |
| Treatment | 2 | 5.03 | 0.0076 |
| Clone | 33 | 4.04 | <0.0001 |
| Error | 458 | | |
| Total | 495 | | |

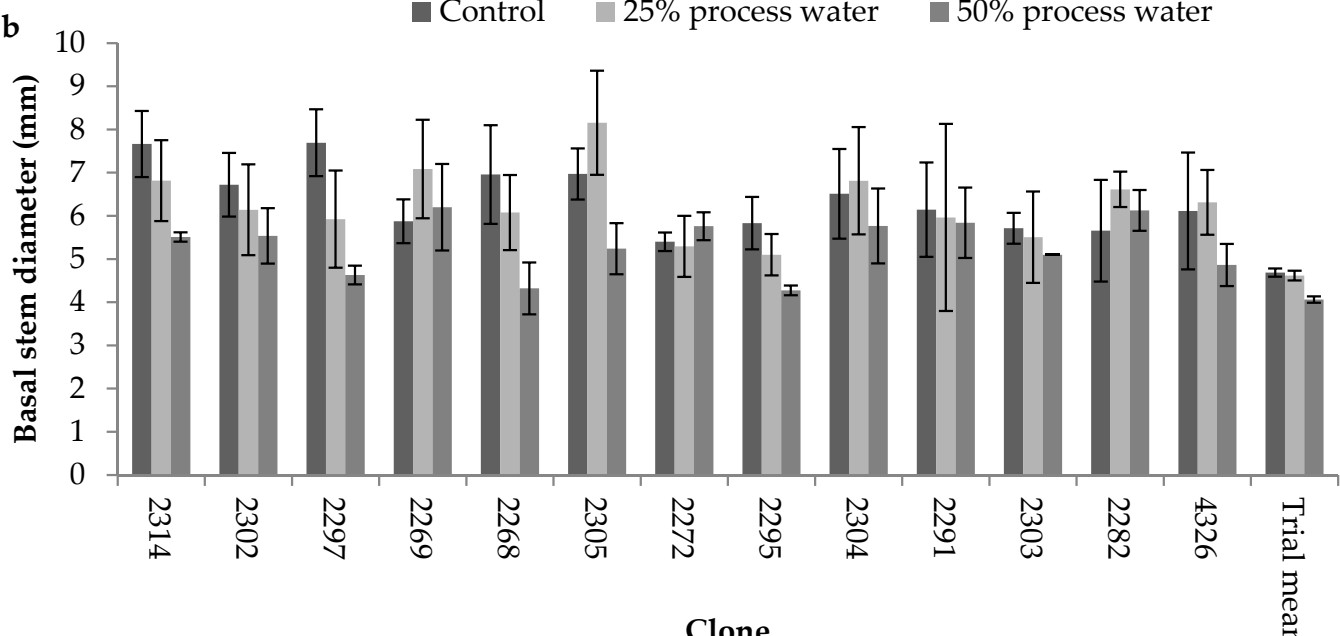

**Figure A1.** (**a**). Mean basal stem basal diameter (±SE, mm) of 12 balsam poplar clones that ranked in the top 30 for all three water treatments as compared with the overall trial mean for 148 clones after 44 days of growth in the 2012 experiment. (**b**). Mean basal stem basal diameter (±SE, mm) of 11 clones that ranked in the top 30 for all three water treatments as compared with the overall trial mean for 86 clones after 65 days of growth in the 2013 experiment.

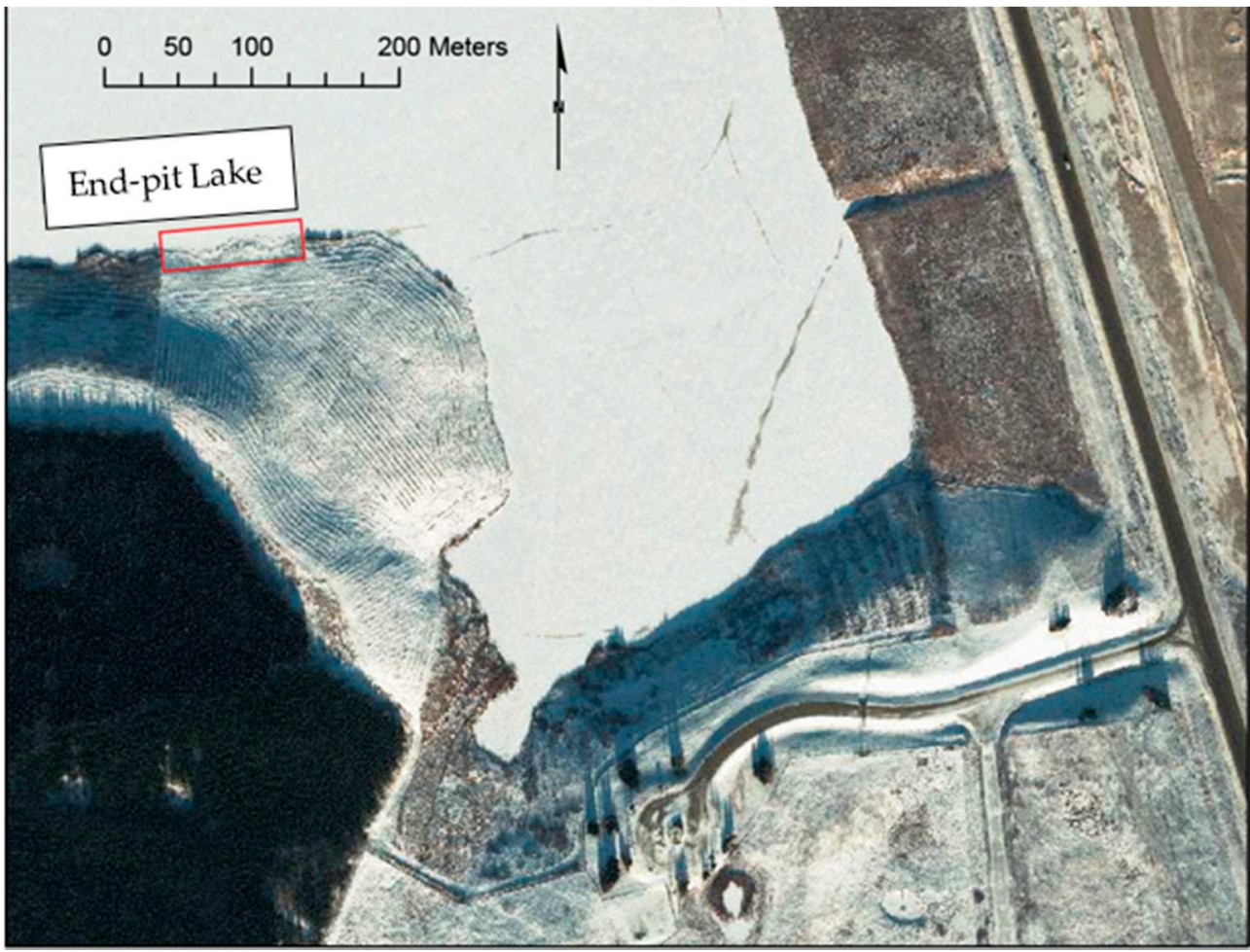

**Figure A2.** Aerial view in winter of the end-pit lake showing the location (marked in red) of the trial adjacent to the lake edge.

**End-pit Lake**

North ↑

**Rep 1**

| 4283 | 2314 | 5578 | 5578 | 947 | 5578 | 3027 | 4643 | 3027 | 2304 | 5578 | 5578 | 4357 | 4304 | 5578 | 5578 | 4285 | 4274 | 2272 | 4296 | 2452 | 2887 | 5578 | 4643 | 4283 | 2453 | 4249 | 4255 | 4293 | 4314 | 3006 | 2997 | 4314 | 4277 | 5578 | 5578 | 3006 | 3006 | 4274 | 3110 |
|---|---|---|---|---|---|---|---|---|---|---|---|---|---|---|---|---|---|---|---|---|---|---|---|---|---|---|---|---|---|---|---|---|---|---|---|---|---|---|---|
| 361 | 362 | 363 | 364 | 365 | 366 | 367 | 368 | 369 | 370 | 380 | 379 | 378 | 377 | 376 | 375 | 374 | 373 | 372 | 371 | 381 | 382 | 383 | 384 | 385 | 386 | 387 | 388 | 389 | 390 | 400 | 399 | 398 | 397 | 396 | 395 | 394 | 393 | 392 | 391 |
| 3188 | 2314 | 4301 | 5578 | 4249 | 2452 | 4295 | 4295 | 947 | 5578 | 3027 | 4304 | 3002 | 4301 | 2997 | 5578 | 2282 | 5578 | 4277 | 3029 | 5578 | 2272 | 2304 | 4277 | 2976 | 5578 | 5578 | 2272 | 5578 | 5578 | 2976 | 4314 | 947 | 5578 | 5578 | 2314 | 5578 | 4357 | 5578 | 5578 |
| 440 | 439 | 438 | 437 | 436 | 435 | 434 | 433 | 432 | 431 | 430 | 429 | 428 | 427 | 426 | 425 | 424 | 423 | 422 | 421 | 411 | 412 | 413 | 414 | 415 | 416 | 417 | 418 | 419 | 420 | 410 | 409 | 408 | 407 | 406 | 405 | 404 | 403 | 402 | 401 |
| 4274 | 2887 | 4296 | 4301 | 3188 | 2995 | 4283 | 3004 | 4357 | 4301 | 3110 | 3002 | 4277 | 4285 | 2272 | 2887 | 4283 | 5578 | 5578 | 4352 | 2976 | 4296 | 2314 | 3110 | 5578 | 4295 | 5578 | 5578 | 5578 | 4326 | 5578 | 3004 | 2452 | 5578 | 4352 | 5578 | 3002 | 5578 | 2995 | 3187 |
| 441 | 442 | 443 | 444 | 445 | 446 | 447 | 448 | 449 | 450 | 451 | 452 | 453 | 454 | 455 | 456 | 457 | 458 | 459 | 460 | 461 | 462 | 463 | 464 | 465 | 466 | 467 | 468 | 469 | 470 | 471 | 472 | 473 | 474 | 475 | 476 | 477 | 478 | 479 | 480 |
| 5578 | 5578 | 2452 | 2453 | 3029 | 3188 | 4304 | 2453 | 3110 | 4285 | 5578 | 3187 | 3029 | 2304 | 5578 | 5578 | 5578 | 4255 | 3004 | 4255 | 2282 | 2997 | 5578 | 2304 | 5578 | 3002 | 4295 | 2997 | 3188 | 5578 | 2282 | 5578 | 4249 | 2453 | 4293 | 4352 | 5578 | 5578 | 5578 | 5578 |
| 520 | 519 | 518 | 517 | 516 | 515 | 514 | 513 | 512 | 511 | 510 | 509 | 508 | 507 | 506 | 505 | 504 | 503 | 502 | 501 | 500 | 499 | 498 | 497 | 496 | 495 | 494 | 493 | 492 | 491 | 490 | 489 | 488 | 487 | 486 | 485 | 484 | 483 | 482 | 481 |
| 5578 | 5578 | 3187 | 3027 | 5578 | 4293 | 4285 | 5578 | 4249 | 4326 | 4274 | 3029 | 4296 | 947 | 2995 | 4643 | 4293 | 5578 | 4352 | 2887 | 3187 | 5578 | 4255 | 5578 | 4326 | 5578 | 4326 | 3004 | 3006 | 4314 | 2995 | 4643 | 2976 | 4357 | 2282 | 5578 | 5578 | 5578 | 4304 | 5578 |
| 521 | 522 | 523 | 524 | 525 | 526 | 527 | 528 | 529 | 530 | 531 | 532 | 533 | 534 | 535 | 536 | 537 | 538 | 539 | 540 | 541 | 542 | 543 | 544 | 545 | 546 | 547 | 548 | 549 | 550 | 551 | 552 | 553 | 554 | 555 | 556 | 557 | 558 | 559 | 560 |

**Rep 2**

| 5578 | 2887 | 4301 | 4293 | 4249 | 5578 | 4326 | 3029 | 4296 | 2314 | 3187 | 5578 | 3188 | 4357 | 5578 | 5578 | 4283 | 2453 | 5578 | 2997 | 4301 | 4352 | 3006 | 3029 | 2282 | 5578 | 2452 | 3002 | 4326 | 2453 | 4357 | 2995 | 5578 | 4301 | 3110 | 5578 | 4296 | 2997 | 4274 | 5578 |
|---|---|---|---|---|---|---|---|---|---|---|---|---|---|---|---|---|---|---|---|---|---|---|---|---|---|---|---|---|---|---|---|---|---|---|---|---|---|---|---|
| 561 | 562 | 563 | 564 | 565 | 566 | 567 | 568 | 569 | 570 | 571 | 572 | 573 | 574 | 575 | 576 | 577 | 578 | 579 | 580 | 581 | 582 | 583 | 584 | 585 | 586 | 587 | 588 | 589 | 590 | 591 | 592 | 593 | 594 | 595 | 596 | 597 | 598 | 599 | 600 |
| 2995 | 5578 | 2997 | 3027 | 5578 | 5578 | 5578 | 5578 | 4314 | 5578 | 5578 | 2272 | 4277 | 2314 | 2452 | 2452 | 5578 | 5578 | 5578 | 2453 | 4643 | 3027 | 2304 | 5578 | 3188 | 3006 | 4255 | 5578 | 5578 | 5578 | 5578 | 4296 | 5578 | 4249 | 3004 | 2976 | 5578 | 4277 | 5578 | 5578 |
| 640 | 639 | 638 | 637 | 636 | 635 | 634 | 633 | 632 | 631 | 630 | 629 | 628 | 627 | 626 | 625 | 624 | 623 | 622 | 621 | 620 | 619 | 618 | 617 | 616 | 615 | 614 | 613 | 612 | 611 | 610 | 609 | 608 | 607 | 606 | 605 | 604 | 603 | 602 | 601 |
| 5578 | 4295 | 5578 | 4326 | 2272 | 4643 | 5578 | 4301 | 3004 | 4283 | 2997 | 4255 | 5578 | 2304 | 5578 | 3006 | 3002 | 4293 | 4274 | 5578 | 2995 | 2314 | 2453 | 4285 | 5578 | 2887 | 4255 | 4304 | 5578 | 4296 | 4277 | 4643 | 4314 | 2314 | 947 | 2976 | 4357 | 4314 | 5578 | 5578 |
| 641 | 642 | 643 | 644 | 645 | 646 | 647 | 648 | 649 | 650 | 651 | 652 | 653 | 654 | 655 | 656 | 657 | 658 | 659 | 660 | 661 | 662 | 663 | 664 | 665 | 666 | 667 | 668 | 669 | 670 | 671 | 672 | 673 | 674 | 675 | 676 | 677 | 678 | 679 | 680 |
| 4352 | 4643 | 4314 | 2304 | 4285 | 5578 | 3110 | 4274 | 4285 | 4249 | 3029 | 4283 | 5578 | 4295 | 4357 | 3027 | 4293 | 5578 | 4274 | 5578 | 4352 | 4304 | 4283 | 4326 | 5578 | 5578 | 2272 | 4304 | 2976 | 2976 | 5578 | 2272 | 2282 | 3187 | 947 | 5578 | 4295 | 3002 | 3110 | 5578 |
| 720 | 719 | 718 | 717 | 716 | 715 | 714 | 713 | 712 | 711 | 710 | 709 | 708 | 707 | 706 | 705 | 704 | 703 | 702 | 701 | 700 | 699 | 698 | 697 | 696 | 695 | 694 | 693 | 692 | 691 | 690 | 689 | 688 | 687 | 686 | 685 | 684 | 683 | 682 | 681 |
| 5578 | 4352 | 2282 | 4285 | 2304 | 5578 | 5578 | 5578 | 2452 | 947 | 3002 | 3110 | 4304 | 4277 | 3004 | 2995 | 2887 | 2282 | 4295 | 4249 | 3188 | 3187 | 3029 | 4293 | 5578 | 5578 | 4255 | 3187 | 3006 | 3027 | 2887 | 5578 | 3188 | 5578 | 5578 | 5578 | 5578 | 947 | 3004 |  |
| 721 | 722 | 723 | 724 | 725 | 726 | 727 | 728 | 729 | 730 | 731 | 732 | 733 | 734 | 735 | 736 | 737 | 738 | 739 | 740 | 741 | 742 | 743 | 744 | 745 | 746 | 747 | 748 | 749 | 750 | 751 | 752 | 753 | 754 | 755 | 756 | 757 | 758 | 759 | 760 |

**Rep 3**

| 5578 | 4357 | 5578 | 4304 | 5578 | 4293 | 4285 | 4285 | 3188 | 2304 | 4357 | 5578 | 5578 | 4277 | 4301 | 3110 | 4301 | 4293 | 5578 | 2887 | 5578 | 2314 | 5578 | 4295 | 5578 | 5578 | 4296 | 2453 | 4295 | 5578 | 3002 | 5578 | 4314 | 3002 | 5578 | 3004 | 5578 | 2997 | 4643 | 3029 |
|---|---|---|---|---|---|---|---|---|---|---|---|---|---|---|---|---|---|---|---|---|---|---|---|---|---|---|---|---|---|---|---|---|---|---|---|---|---|---|---|
| 761 | 762 | 763 | 764 | 765 | 766 | 767 | 768 | 769 | 770 | 771 | 772 | 773 | 774 | 775 | 776 | 777 | 778 | 779 | 780 | 781 | 782 | 783 | 784 | 785 | 786 | 787 | 788 | 789 | 790 | 791 | 792 | 793 | 794 | 795 | 796 | 797 | 798 | 799 | 800 |
| 4277 | 4283 | 4285 | 3188 | 4249 | 4296 | 5578 | 4352 | 2995 | 4357 | 3002 | 4314 | 2314 | 2997 | 4304 | 2452 | 4352 | 4643 | 3029 | 5578 | 5578 | 4352 | 4274 | 3006 | 947 | 5578 | 4357 | 2453 | 3110 | 4296 | 5578 | 2452 | 2282 | 5578 | 5578 | 4285 | 5578 | 5578 | 4296 | 2272 |
| 840 | 839 | 838 | 837 | 836 | 835 | 834 | 833 | 832 | 831 | 830 | 829 | 828 | 827 | 826 | 825 | 824 | 823 | 822 | 821 | 820 | 819 | 818 | 817 | 816 | 815 | 814 | 813 | 812 | 811 | 810 | 809 | 808 | 807 | 806 | 805 | 804 | 803 | 802 | 801 |
| 4304 | 5578 | 5578 | 4283 | 5578 | 4295 | 3110 | 5578 | 3004 | 3006 | 2995 | 947 | 5578 | 4277 | 3027 | 2976 | 5578 | 5578 | 5578 | 2887 | 5578 | 947 | 2453 | 2452 | 2272 | 5578 | 4326 | 5578 | 4326 | 2282 | 5578 | 4255 | 5578 | 4326 | 5578 | 2314 | 3188 | 2887 | 4314 |  |
| 841 | 842 | 843 | 844 | 845 | 846 | 847 | 848 | 849 | 850 | 851 | 852 | 853 | 854 | 855 | 856 | 857 | 858 | 859 | 860 | 861 | 862 | 863 | 864 | 865 | 866 | 867 | 868 | 869 | 870 | 871 | 872 | 873 | 874 | 875 | 876 | 877 | 878 | 879 | 880 |
| 5578 | 5578 | 2997 | 5578 | 5578 | 4283 | 5578 | 3187 | 5578 | 5578 | 2976 | 2997 | 3027 | 5578 | 2453 | 2304 | 2995 | 2282 | 2282 | 5578 | 4274 | 4249 | 4326 | 3029 | 2887 | 5578 | 3187 | 4643 | 5578 | 2272 | 4293 | 3006 | 3027 | 4643 | 5578 | 4314 | 5578 | 5578 | 5578 | 2272 |
| 920 | 919 | 918 | 917 | 916 | 915 | 914 | 913 | 912 | 911 | 910 | 909 | 908 | 907 | 906 | 905 | 904 | 903 | 902 | 901 | 900 | 899 | 898 | 897 | 896 | 895 | 894 | 893 | 892 | 891 | 890 | 889 | 888 | 887 | 886 | 885 | 884 | 883 | 882 | 881 |
| 4249 | 4301 | 2976 | 3187 | 3006 | 5578 | 2314 | 2976 | 4283 | 5578 | 5578 | 4352 | 2995 | 4295 | 3029 | 5578 | 3187 | 4301 | 4304 | 4274 | 3004 | 4274 | 947 | 4249 | 3004 | 3002 | 5578 | 5578 | 4277 | 2304 | 3110 | 5578 | 3188 | 5578 | 4293 | 2452 | 4255 | 5578 | 4255 | 2304 |
| 921 | 922 | 923 | 924 | 925 | 926 | 927 | 928 | 929 | 930 | 931 | 932 | 933 | 934 | 935 | 936 | 937 | 938 | 939 | 940 | 941 | 942 | 943 | 944 | 945 | 946 | 947 | 948 | 949 | 950 | 951 | 952 | 953 | 954 | 955 | 956 | 957 | 958 | 959 | 960 |

| XXXX | ← Clone Number |
|---|---|
| xxx | ← Position Number |

**Treatment 1 (25 selected tolerant clones):** 2272, 2282, 2304, 2314, 2453, 2976, 2995, 2997, 3027, 3029, 3110, 3187, 3188, 4249, 4255, 4274, 4277, 4283, 4285, 4295, 4296, 4301, 4304, 4326, 4357

**Treatment 2 (10 selected control clones):** 947, 2452, 2887, 3002, 3004, 3006, 4352, 4293, 4314, 4643

**Treatment 3 (Stream 1 vegetative lot clones):** 5578

**Figure A3.** Map of the field trial site showing individual tree locations within each block (rep).

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
