# Peer review of "Field Testing of Selected Salt-Tolerant Screened Balsam Poplar (Populus balsamifera L.) Clones for Use in Reclamation around End-Pit Lakes Associated with Bitumen Extraction in Northern Alberta"

_forests, doi:10.3390/f12050572_

Round 1
Reviewer 1 Report
This manuscript is a very detailed description of both greenhouse and field testing of balsam poplar clones for establishment and reclamation of highly-saline soils in northern Alberta. The topic fits very well into Forests, and this information will help to advance the use of poplars (and other short rotation woody crops) for phytotechnologies. The authors did a masterful job designing both studies and an even better job conducting and reporting on their results. In fact, this is one of the best manuscripts I have reviewed in a very long time. As such I do not have any major concerns or comments. Some minor suggestions are listed below. I thank the authors for producing such a high-quality paper. Great job!
*Title: Populus balsamifera L. (please add L.)
*Materials/Methods: You probably thought about this, but a map showing the field site, nursery, etc. could be helpful for an international readership.
*L110: 10-cm
*L140 (and throughout the paper): Be sure to use mathematical multiplication symbols rather than the letter "x", where appropriate (e.g., also on L257).
*L198: stuck
*L211: A figure showing the field layout is not necessary but could be meaningful.
*L266: Adding an ANOVA P-value table to the appendices could be useful.
*Table 1 caption: Define SNK
*Table 1: "Measurements" headings are not needed; Treatment rather than Treatments
*L319: Pearson's correlation coefficient
*Table 2 caption: "...25% process water (Process..."
*Figure 1 (and throughout): Have you considered conducting analyses of means in SAS? They could be useful for comparing your clones to the trial mean.
*Table 3 caption: among treatments
*Figure 4: Excellent presentation of the data over time
*L454: height nor DBH
*Figure 8 caption: ...trees planted in 2019 (1 = 25...)
*Table 5 footnote: Delete "in this figure"
*Table 7 caption: two-year-old
*L514: results suggest that
*L517 (and throughout): clone designations should be bookended by single parens 'AP4357'
*L517: OSPW-tolerant
Reviewer 2 Report
Comments and Suggestions for the Authors:
Field testing of selected salt tolerant screened balsam poplar (Populus balsamifera) clones for use in reclamation around end-pit lakes associated with bitumen extraction in northern Alberta
Manuscript ID: forests-1179753
A brief summary
The subject of the manuscript is interesting, fitting well in the scope of the Forests Journal. The issues taken up by the authors are very topical in the context of the salinity contamination and its reduction. It is a very interesting and helpful study applicable for the restoration of saline in the contaminated areas.
However it would be useful to describe better the research field area. Could you, please, provide the field area characterisation (map of the study field and the individual plants distribution)?
The tolerance of plants to salinization as well as alkalinisation is different. Crop tolerance also depends on climate factors (temperature, humidity, sunshine intensity, humidity, etc.). What were the climatic conditions during the compared years? Could they affect the presented research?
Plants tolerance for salt content is given by two coefficients: 1.) soil salinity limit, 2.) the yield reduction gradient, which is an expression of the percentage reduction in yield per unit of increased salinity above a certain limit. Could you please indicate these values during the individual compared years?
Specific comments
Line 71: I suggest to use the same formatting for numbers (thousands separation): 8000 mg L-1, but e.g. in the line 33 it is 56,000 barrels of bitumen per day, but also elsewhere.
Line 347: Where is Fig. A1a) and Fig. A1b
Figure 1: Please mark the units on the x-axis to be more comprehensive for readers.
Figure 2: I recommend applying the same scale for the y-axis so that the values can be better compared.
